materials science

graphite, ultrasound, flotation surface properties

**Author for correspondence:**
Wenze Kang
e-mail: kwz010@163.com

# Enhancement of flaky graphite cleaning by ultrasonic treatment

## Wenze Kang and Huijian Li

Institute of Mining Engineering, Heilongjiang University of Science and Technology, Harbin 150022, China

WK, 0000-0002-4696-5890

In this study, the aim is to simplify the graphite cleaning process. In order to achieve flotation for graphite effectively, ultrasonic treatment was used as a pre-treatment technique. Flotation tests were conducted using different ultrasound power and ultrasonic treatment time. The influences of ultrasonic treatment on particle sizes, morphologies, wettability, the content of surface elements and on the flotation effect of flaky graphite were investigated. The results of ultrasonic treatment for graphite flotation were compared with the results of conventional flotation. The results showed that ultrasonic treatment not only changed the size of flaky graphite, but also eliminated impurities on the graphite surface. Additionally, the ultrasonic treatment improved the hydrophobicity of graphite. It was observed that ultrasound can remove not only silicate impurities but also most other metal impurities. The yield, carbon content and recovery of flotation concentrate were 91.46%, 95.17% and 96.12% after ultrasonic treatment for 4 min with ultrasound power 1600 W, which were 5.83%, 2.86% and 8.84% higher than that of conventional flotation, respectively. The graphite after ultrasonic treatment was conducted only one times flotation, the carbon content in concentrate products had reached 95%. This study indicates that intensifying graphite flotation by ultrasonic treatment can shorten the graphite cleaning process.

## 1. Introduction

Flotation is one of the main methods for separating fine-grained minerals. Owing to there being different types of fine-grained minerals, the process of flotation also varies. Generally, it can satisfy the requirement of separating fine coal only through one flotation cycle, however, for flotation of graphite, one flotation operation cannot meet industrial requirements [1–3]. Owing to the low grade of raw graphite (generally, in the range of carbon content 5–15%) and high requirement imposed upon the final grade of concentrates (a carbon content of greater than 95%), the

**Figure 1.** Technological processing flow-sheet of the Luobei graphite concentrator.

required purity of the concentrate cannot be reached through one flotation operation. The requirement cannot be met unless multiple grinding and flotation processes are carried out. For example, in a certain graphite concentrator in Luobei County, Heilongjiang Province, China, the grade of raw graphite was 10.38%, and the carbon content in graphite cannot reach 95% before being subjected to crushing, coarse and rougher, nine stages regrinding and 10 stages cleaning. Technological process flow-sheet of the graphite concentrator in Heilongjiang, China is shown in figure 1. In the cleaning process, on condition that the carbon content in graphite rose to 90%, it was necessary to conduct one ore-grinding step and one cleaning step each time a 1% increase in carbon content in graphite was needed: this entailed a long flotation process and high energy consumption. Therefore, it is necessary to intensify the cleaning of graphite by selecting a technological method with which to shorten the flotation process for graphite.

In recent years, researches into ultrasonic treatment of intensifying mineral flotation have gradually increased. Many scholars have explored the flotation process of oxidized coal and coal with a high ash content using ultrasonic treatment. Existing studies showed that, at the same dosage, a higher yield of clean coal and lower ash content in the clean coal can be attained by conducting ultrasonic treatment flotation [4–11]. Some researchers investigated the influence of ultrasonic treatment on the flotation of high-sulfur coal to reveal that ultrasonic treatment can promote the dissociation between coal and pyrite and increase the difference of floatability between coal and pyrite. The ultrasonic treatment flotation for high-sulfur coal showed remarkable de-ashing and de-sulfurizing effects [12–22]. Some other researchers studied the influence of ultrasonic treatment on the flotation of borocalcite–clay, magnesite–quartz, lead sulfide–xanthate, fluorite–quartz, barite–copper pyrites and borate–josephinite. Their researches indicated that ultrasonic treatment can selectively change the surface properties of

targeted mineral particles, enhance the selectivity of flotation and improve the flotation efficiency [23–27]. Letmathe *et al.* [28] investigated the flotation process of graphite and reported the influence of ultrasonic treatment on the flotation effect of graphite. Ultrasonic treatment can improve the effectiveness of a reagent due to a more uniform distribution in the suspension. Barma *et al.* [29] investigated the feasibility of low-frequency ultrasound in enhancing the floatability of flaky graphite from low-grade graphite ore. Their results showed that the yield, fixed carbon content and percentage recovery of the flotation concentrate products increased significantly under ultrasonic-assisted flotation.

Previous researchers have made remarkable achievements in enhancing minerals flotation by ultrasound [21,30], but the effect of ultrasonic treatment on graphite cleaning process is studied less [29]. To develop a method for shortening the graphite cleaning process, the influences of ultrasonic treatment on changes of particle sizes, morphologies, wettability, content of surface elements of flaky graphite and on the flotation effect of flaky graphite were explored in this paper. The emphasis of this paper is on the enhancement of graphite flotation by ultrasound. The aim is to simplify the graphite cleaning process and promote the efficient development of the graphite industry.

# 2. Test materials and methods

## 2.1. Samples

The test samples were taken from a graphite concentrator in Luobei, Heilongjiang Province, China. The beneficiation process of the concentrator consisted of crushing, coarse and rougher, nine stages regrinding and 10 stages cleaning as shown in figure 1. The current samples were taken in the concentrate tank after the sixth flotation phase. The samples were collected with a sampling bucket (1 l volume) at 15 min intervals. After several consecutive samplings, the samples were aggregated, dewatered, dried and then bagged. A total of 30 kg sample was obtained and was used for conducting the present study. The average carbon content of the samples was 90.56%.

## 2.2. Test on flotation and ultrasonic treatment

An XFD-1.0 flotation machine was employed, with the volume of the flotation cell of 1 l. Kerosene was used as a collector while secondary octyl alcohol served as a frother. By selecting three main influencing factors (the pulp density and the dosages of kerosene and secondary octyl alcohol), the optimal parameters (involving pulp density and dosages of kerosene and secondary octyl alcohol of $40 \text{ g l}^{-1}$, $300 \text{ g t}^{-1}$ and $125 \text{ g t}^{-1}$, respectively) for flotation were determined through orthogonal experiment.

In the ultrasonic experiments, a JAC-5500 ultrasonic generator was employed, working at the frequency of 25 kHz and output power of up to 2 kW. The samples were ultrasonically treated at 800, 1200 and 1600 W, for 2, 3, 4, 5 and 6 min, respectively.

In the experiments, 40 g graphite was placed into a beaker, to which 500 ml water was added. The mixture was stirred and then ultrasonically treated. The ultrasonic treatment increased the pulp temperature, which affects the flotation process in many ways. For instance, it can increase the activity of amine molecules [31]. In order to eliminate the influence of temperature on graphite flotation, the ultrasonic treatment pulp was cooled to room temperature. Subsequently, the cooled pulp was put into the flotation machine and stirred for 2 min. Kerosene was added and the mixture stirred for a further minute. This was followed by the addition of secondary octyl alcohol into the mixture, stirring for 30 s and flotation for 2 min. Next, concentrates and tailings were collected, filtered and dried separately. The dried samples were weighed to calculate the yield of the products. By using high-temperature burning, the ash content and volatile matter in graphite were measured according to Chinese Standard GB T 3521–2008.

The carbon content of the flotation products is given by

$$\beta = 100 - A_d - V_d, \tag{2.1}$$

where $\beta$, $A_d$ and $V_d$ represent the carbon content in products (%), the ash content (%) and volatile matter (%), respectively.

The recovery of products is given by

$$\varepsilon = \left( \frac{\beta \times \gamma_j}{\alpha} \right) \times 100\%, \tag{2.2}$$

where $\varepsilon$, $\alpha$, $\gamma_j$ and $\beta$ represent recovery (%), carbon content in original samples (%), yield of products (%) and carbon content in products (%), respectively.

Forty grams of graphite was uniformly mixed with 500 ml of water, which was ultrasonically treated for 5 min and then filtered and dried. Afterwards, the graphite with the same mass was taken and uniformly mixed with 500 ml of water and the mixture was filtered and dried. By using an MX2600 scanning electron microscope (SEM), the surface morphologies of graphite, in ultrasonicated and untreated states were separately detected.

Forty grams of graphite was uniformly mixed with 500 ml of water and then the mixture was ultrasonically treated for 5 min. Afterwards, wet screening was carried out. Graphite with the same mass, but untreated, was also wet screened.

# 3. Test results and discussion

## 3.1. The influence of ultrasonic treatment at different powers on the flotation effect of flaky graphite

According to §2.2, flotation tests were conducted by conventional flotation and ultrasonic-assisted flotation process at different ultrasound power (800, 1200 and 1600 W) and different ultrasonic treatment time separately. The yield, carbon content and recovery of flotation concentrate products were obtained during conventional and ultrasonic-assisted flotation process at different power and different ultrasonic treatment time. The results are presented in figure 2.

Points of the ultrasonic treatment time 0 min in figure 2a–c represent the products parameters obtained by conventional flotation. The yield, carbon content and recovery of the concentrate products of conventional flotation are 85.63%, 92.31% and 87.28%, respectively. It can be seen from figure 2 that the yield, carbon content and recovery of concentrate products obtained during ultrasonic-assisted flotation are higher in comparison with the concentrate products of conventional flotation. For example, the yield, carbon content and recovery of flotation concentrate were 91.46%, 95.17% and 96.12% after ultrasonic treatment for 4 min with ultrasonic power 1600 W, which were 5.83%, 2.86% and 8.84% higher than that of conventional flotation, respectively. On the condition that the ultrasonic treatment was conducted for longer than 4 min, the carbon content of concentrate products exceeds 95%. At 800, 1200 and 1600 W, with the increase of ultrasonic power, the carbon content of concentrates increased slightly while the yield and recovery of the concentrates both increased significantly. The most optimal flotation effect could be obtained when the ultrasonic power was 1600 W. Through the above analysis, the carbon content of the flotation concentrate products reached 95% after only one flotation step under ultrasonic-assisted flotation. This test verified that the introduction of ultrasonic-assisted flotation can simplify the cleaning process of graphite. Technological graphite flotation processing flow-sheet using ultrasonic treatment is shown in figure 3. Owing to the optimal flotation effect being found at an ultrasonic power of 1600 W, the power of 1600 W was used in the following ultrasonic treatment experiments.

## 3.2. The influences of ultrasonic treatment on particle size and morphologies of flaky graphite

### 3.2.1. The influence of ultrasonic treatment on morphologies of flaky graphite

The morphology of graphite surface was examined by scanning electron microscopy (SEM), as shown in figure 4.

Figure 4a is the SEM image of raw graphite, and figure 4b is the SEM image of graphite after ultrasonic treatment for 5 min. Figure 4a shows that there are small pieces of bright components in the flaky graphite surface, which are associated minerals and fine mud adhering to the graphite surface. As shown in figure 4b, the fine bright components were significantly reduced, implying that the associated minerals or fine mud adsorbed onto the graphite surface were eliminated after ultrasonic treatment. By conducting ultrasonic treatment, the impurities on the graphite surface were cleaned off to expose a fresh surface of the graphite [32,33]. It can be supposed that ultrasonic treatment may increase the carbon content of graphite and enhance the hydrophobicity of graphite.

### 3.2.2. The influence of ultrasonic treatment on particle size of flaky graphite

To reveal the effects of ultrasonic treatment on graphite particle size, screening tests were carried out on original and ultrasonically pre-treated graphite samples separately. Test results of graphite screening are shown in table 1. It can be seen from table 1, for the particle-size fractions of greater than 0.25, 0.25–0.15

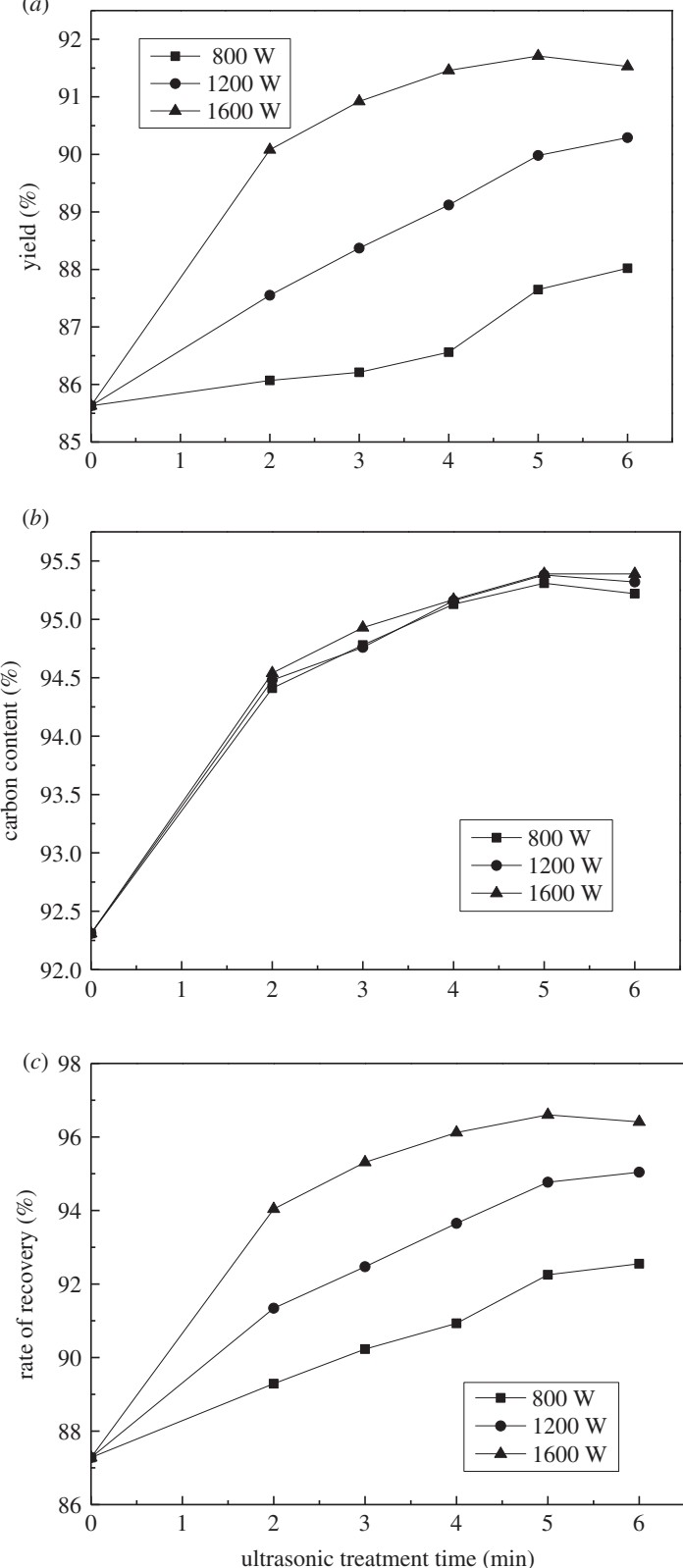

**Figure 2.** Effect of graphite flotation at different ultrasonic power. (*a*) Yield of concentrates of graphite flotation at different ultrasonic power. (*b*) Carbon content of concentrates of graphite flotation at different ultrasonic power. (*c*) Recovery of concentrates of graphite flotation at different ultrasonic power.

and 0.15–0.074 mm, the yields of graphite at different fractions after ultrasonic treatment were lower than those without ultrasonic treatment. This indicated that ultrasonic treatment showed a significant breaking effect on larger-sized graphite particles [34]. For fractions 0.074–0.045 and less than

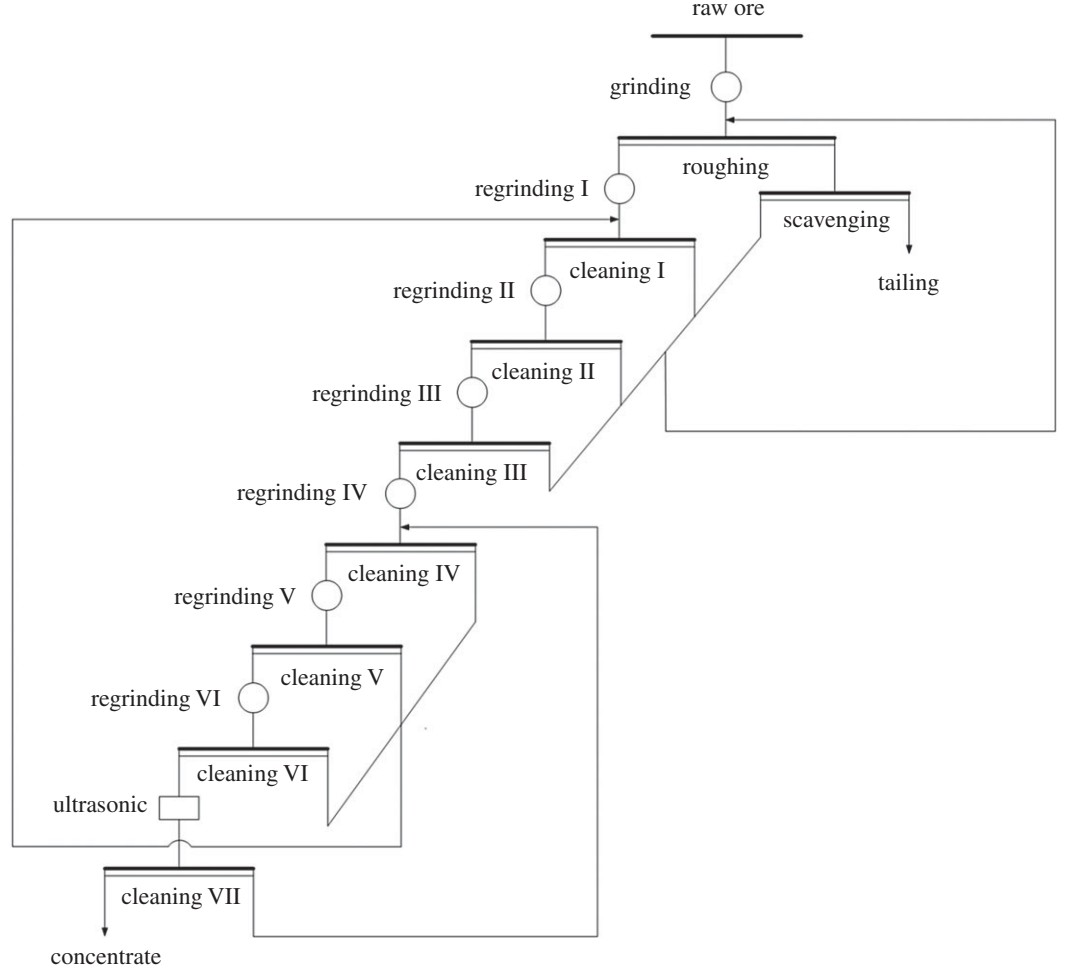

**Figure 3.** Technological graphite flotation processing flow-sheet using ultrasonic treatment.

0.045 mm, the yields of graphite at various fractions undergoing ultrasonic treatment were higher than those without ultrasonic treatment. It was possible that the graphite at the fractions of greater than 0.25, 0.25–0.15 and 0.15–0.074 mm was broken to that at fractions of 0.074–0.045 and less than 0.045 mm, thus, the yields of graphite with fractions of 0.074–0.045 and less than 0.045 mm increased. Ultrasound can also break graphite at the fractions of 0.074–0.045 and less than 0.045 mm, however, owing to the graphite at these fractions being too small and some broken fine particles of graphite at the other fractions interfered with the results, this failed to reflect the breaking effect of ultrasound on graphite at these two fractions. The carbon contents in graphite at five fractions all increased after ultrasonic treatment and ultrasonic treatment can eliminate impurities on the graphite surface. In the filtering process, fine impurities were adsorbed onto filter paper, thus resulting in the increase of carbon content in graphite. Based on the yields and carbon contents at different particle sizes in graphite being subjected to ultrasonic treatment, it can be seen that ultrasonic treatment showed a certain breaking and cleaning effect on graphite.

## 3.3. The influence of ultrasonic treatment on the wettability of flaky graphite

### 3.3.1. The influence of ultrasonic treatment on the contact angle of flaky graphite

In the experiments, based on a direct method, the contact angle of graphite was measured using an XG-CAMA1 enhanced static contact goniometer. Six graphite powder samples were prepared, one of which was not treated by ultrasound and the other five samples were treated by ultrasound for 2, 3, 4, 5 and 6 min respectively. Two grams of each graphite powder sample was put in a 40 mm diameter mould each time and pressed to form pellets at a constant pressure of 10 MPa and continuous pressurization for 2 min. A smoother area was selected on the surface of the pellets. By using a

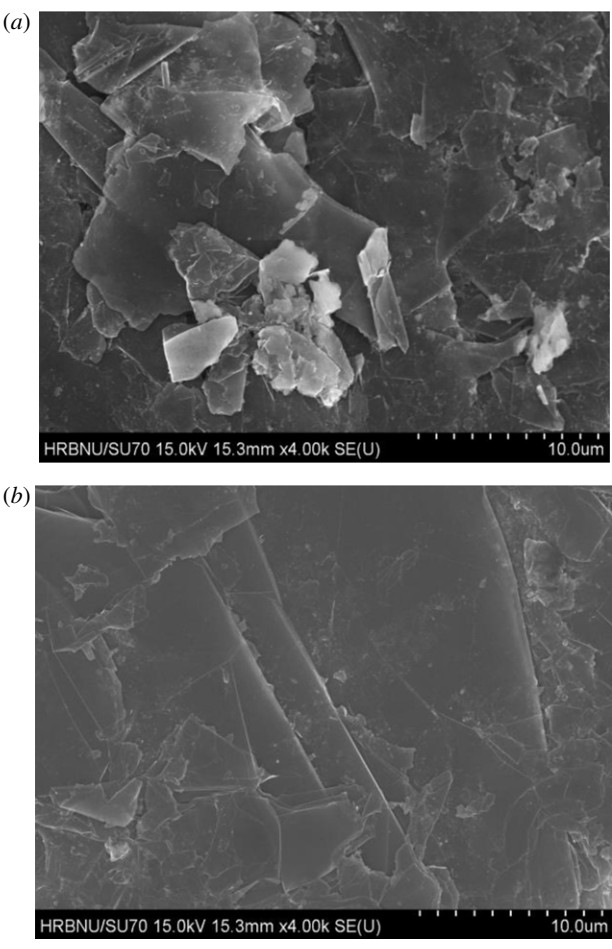

**Figure 4.** Surface morphology of graphite (*a*) without ultrasonic treatment and (*b*) with ultrasonic treatment.

**Table 1.** Test results of graphite screening (%).

| particle-size fraction (mm) | original graphite | | ultrasonic treatment graphite | | fraction change rate | |
|---|---|---|---|---|---|---|
| | yield | carbon content | yield | carbon content | yield | carbon content |
| >0.25 | 1.44 | 94.76 | 0.77 | 95.60 | −46.53 | 0.89 |
| 0.25–0.15 | 9.71 | 94.15 | 3.59 | 95.27 | −63.03 | 1.19 |
| 0.15–0.074 | 37.32 | 92.93 | 20.19 | 94.35 | −45.90 | 1.54 |
| 0.074–0.045 | 8.34 | 91.55 | 8.59 | 92.38 | 3.00 | 0.91 |
| <0.045 | 43.19 | 89.14 | 66.85 | 90.43 | 54.78 | 1.45 |

microlitre syringe, the test solution (distilled water) was placed onto the graphite surface to form hemispherical drops, each with a diameter of about 1 mm and the drops photographed. According to the height of the drops and the radius of the bottom surface, the contact angle can be calculated. Three points were selected from each pellet to measure the corresponding contact angle and the average value of three measurements was taken. The measurement results are shown in table 2.

In this study, the contact angle of the ultrasonic treatment graphite was compared with that of the raw graphite. Table 2 shows when the ultrasonic treatment time is 0 min, i.e. without ultrasonic treatment, the contact angle of the raw graphite pellet is 86.37°. It can be seen from table 2 that with increasing ultrasonic treatment time, the contact angle between graphite and water gradually increased. After longer than 5 min, the contact angle did not change to any further significant extent. Ultrasonic treatment caused an increase in contact angle between graphite and water, which implied that the

**Table 2.** Contact angle measured after graphite pressed into pellets.

| ultrasonic treatment time (min) | 0 | 2 | 3 | 4 | 5 | 6 |
|---|---|---|---|---|---|---|
| contact angle (°) | 86.37 | 87.15 | 88.11 | 89.34 | 89.61 | 89.56 |

**Table 3.** The wetting heat of graphite and kerosene at different particle size (J g$^{-1}$).

| particle-size fractions (mm) | original graphite | ultrasound-treated graphite |
|---|---|---|
| 0.25–0.15 | 0.071 | 0.148 |
| 0.15–0.74 | 0.083 | 0.300 |
| 0.074–0.045 | 0.307 | 0.326 |
| <0.045 | 0.489 | 0.561 |

ultrasonic treatment strengthened the hydrophobicity of graphite. The results can be explained by the effect of acoustic cavitation that cleans particle surfaces [30,35,36].

### 3.3.2. The influence of ultrasonic treatment on the wetting heat of flaky graphite

Generally, water is used as the wetting liquid to measure the wetting heat of minerals. The flaky graphite shows a low density and a strong hydrophobicity and is flaky so it cannot be completely wetted by water. Therefore, the wetting heat cannot be accurately measured by using water as the wetting liquid. Through testing, it can be found that kerosene can wet flaky graphite and thus kerosene was selected as the wetting liquid.

By using a Setaram C80D microcalorimeter, the wetting heat of graphite was measured. In each test, 0.5 g of graphite was weighed and 2 ml of kerosene (the wetting liquid) was taken. The samples were kept at the initial temperature of 26°C for 2 h during the test and the data on wetting heat were collected over a 3 h period.

The prepared graphite samples were ultrasonically treated for 5 min. The ultrasonic treatment can change the particle size of graphite. In order to eliminate the influence of particle size change on the results of wetting heat, screening tests were carried out. To reveal the effects of ultrasonic treatment, the screening tests were conducted on conventionally and ultrasonically pre-treated graphite samples, respectively. As a result, the graphite was screened into four particle-size fractions (0.25–0.15, 0.15–0.74, 0.074–0.045 and less than 0.045 mm). By comparing the wetting heat in graphite without and with ultrasonic treatment at the same fraction, the influence of ultrasonic treatment on the wettability of graphite can be reflected [37]. The results of the wetting heat of graphite and kerosene at different particle size are shown in table 3.

As shown in table 3, compared with original graphite, the ultrasound-treated graphite at each fraction exhibited an increased wetting heat with kerosene. This indicated that the interaction between ultrasound-treated graphite and kerosene was strengthened. According to the 'like dissolves like' principle, the more similar the properties of a mineral and the wetting liquid, the stronger the interaction between the two parameters and the greater the wetting heat [38]. Kerosene is hydrophobic, so the wetting heat between kerosene and ultrasonically treated graphite was larger than that between kerosene and untreated graphite. This indicated that the surface properties of ultrasonically treated graphite were more similar to those of hydrophobic kerosene. By analysing the changes in wetting heat of ultrasonically treated graphite and original graphite, it can be inferred that the hydrophobicity of graphite increased after ultrasonic treatment. The change regularity of the wetting heat of graphite and kerosene agrees with that of the contact angle of the graphite and water, showing that ultrasonic treatment increases hydrophobicity of graphite.

## 3.4. The influence of ultrasonic treatment on the element content of flaky graphite surface

In this study, the contents of mineral elements were measured using an S4EXPLORER XRF spectrometer. The element content of original samples, concentrates obtained by conventional flotation, ultrasonically

**Table 4.** Measurement results of the main elements in graphite (%).

| element | original sample | concentrate of conventional flotation | ultrasonically treated sample | concentrate of ultrasonic flotation |
|---|---|---|---|---|
| Al | 6.398 | 6.617 | 8.543 | 5.5 |
| Si | 18.109 | 15.652 | 22.06 | 13.895 |
| P | 0.423 | 0.5 | 1.057 | 1.743 |
| S | 7.374 | 5.262 | 5.758 | 6.716 |
| K | 7.541 | 8.48 | 12.497 | 7.424 |
| Ca | 9.085 | 5.33 | 6.9 | 5.23 |
| Fe | 41.749 | 47.051 | 31.51 | 46.327 |
| Cl | 1.639 | 2.003 | 2.313 | 2.639 |
| Ag | 0.781 | 0.942 | 0.541 | 1.473 |

treated samples and concentrates obtained by ultrasonically pre-treated flotation were measured. Measurements results are shown in table 4.

It can be seen from table 4, the elemental Si, S and Ca contents in concentrates obtained by using conventional flotation all decreased compared with the original samples. This indicated that conventional flotation can decrease impurities and increase the carbon content in concentrates to some extent. However, the amounts of other metal elements except Ca increased. This implied that conventional flotation was unsuccessful in reducing the contents of a majority of metal elements and those impurities removed through conventional flotation most likely appeared as silicate minerals. Compared with the original samples, the elemental Fe, Ag, Ca and S contents in ultrasonically treated samples were reduced because impurities (i.e. metal minerals) on the graphite surface were removed by ultrasonic treatment and fine impurities were adsorbed onto the filter and were thereby lost in the filtering process, thus resulting in the decreased amounts of the corresponding elements present on the graphite surface. This was consistent with the test result given in §3.2.2, where the carbon contents in graphite at five particle-size fractions all increased after ultrasonic treatment. Moreover, the elemental S content decreased, which indicated that some of the ferric compounds lost in the process appeared as pyrite. Compared with concentrates obtained using conventional flotation, the elemental contents of Al, Si, K, Ca and Fe of flotation concentrate products obtained during ultrasonic-assisted flotation process decreased. That is, except for Ag, the elemental contents of all other metals decreased. The content of non-metal element Si was reduced, which indicated that ultrasonic treatment flotation not only can eliminate most metal elements, but also can remove silicate impurities. According to the analysis of product elements, it can be verified that the effect of ultrasonic treatment flotation was superior to that of conventional flotation.

# 4. Conclusion

1) Ultrasonic treatment not only changed the size of flaky graphite, but also eliminated impurities on the graphite surface. The carbon content in ultrasonically treated graphite increased compared with that in original graphite. The research showed that ultrasonic treatment exhibited a certain breaking and cleaning effect on flaky graphite.

2) By investigating the contact angle and wetting heat of flaky graphite, it can be seen that, compared with original graphite, ultrasonically treated graphite exhibited improved hydrophobicity, so ultrasonic treatment was conducive to flotation of graphite.

3) By exploring the elemental composition of the flaky graphite surface, it can be found that the Si, S and Ca contents in graphite decreased after conventional flotation. This indicated that conventional flotation can remove silicate impurities and a small number of metal impurities. Compared with conventional flotation, the elemental Al, Si, K, Ca and Fe contents in graphite decreased after ultrasonic treatment flotation. This implied that ultrasonic treatment flotation can not only eliminate silicate impurities, but also remove most metal impurities. Based on an elemental analysis of the resulting concentrates, the effect of ultrasonic treatment flotation was shown to be superior to that of conventional flotation.

4) The flotation tests on graphite verified that the yield, carbon content and recovery of concentrates obtained through ultrasonic treatment flotation were much greater than those attained using conventional flotation. The carbon content of the flotation concentrates products reached 95% after only one flotation step under ultrasonic-assisted flotation. This revealed that, after ultrasonic treatment, the carbon content in concentrates can satisfy the requirement for cleaning of graphite by only one flotation step. The research validated the idea that intensifying the cleaning of graphite by using ultrasound can shorten the cleaning process for graphite.

Data accessibility. Our data are deposited at Dryad Digital Repository: https://doi.org/10.5061/dryad.wm37pvmhf [39].
Competing interests. There are no conflicts to declare.
Funding. This work was financially supported by the National Science & Technology Pillar Program of China (grant no. 2013BAE04B01) and Heilongjiang Provincial Department of Human Resources and Social Security.

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
