## [Reviewer comments · Royal Society Open Science]

Review History

RSOS-191160.R0 (Original submission)

Review form: Reviewer 1

Is the manuscript scientifically sound in its present form?

Yes

Are the interpretations and conclusions justified by the results?

Yes

Is the language acceptable?

Yes

Do you have any ethical concerns with this paper?

No

Have you any concerns about statistical analyses in this paper?

No

Recommendation?

Accept with minor revision (please list in comments)

Comments to the Author(s)

Minor revision is recommended. This manuscript may be accepted after fulfilling the following comments:

- Written English of the manuscript needs to be improved for better clarity.
- Authors may add some lines or texts to the revised manuscript precisely indicating the importance of this research, its novelty and industrial relevance.
- Include these papers in the introduction section to support your work strongly:
DOI: <https://doi.org/10.1016/j.ultsonch.2019.04.033>
DOI: <https://doi.org/10.1016/j.ultsonch.2018.08.016>
DOI: <https://doi.org/10.1021/acs.energyfuels.9b01543>
- Include few more literatures particularly on the work published on flake graphite and discuss how the published work is different than your work. You can use this paper to have some preliminary idea:
DOI: <https://doi.org/10.1016/j.ultsonch.2019.04.033>
- Comparison between conventional and ultrasonicated flotation may be provided using recovery, yield and fixed carbon in the abstract.
- In section 2.2, authors may provide reference against the “orthogonal experiments” for better clarity of the optimization process being used.
- Authors have mentioned that, at higher power (1600 W), yield and recovery were found to be higher than those of 800 and 1200 W. Now the major concern here is its economic aspects since the higher power will consume more energy. Kindly justify this.
- What are the values of yield, recovery and fixed carbon content of conventional flotation? Please compare them for more clarity in section 3.1.
- In section 3.4, the elemental compositions of each feed and products varies differently. For instance, “Al and Fe” in the concentrate of conventional flotation is higher than its original sample. Similar trends can also be seen in concentrate of ultrasonicated flotation. Kindly comment on this.
- Contact angle on ultrasonicated graphite concentrate may be compared with those without ultrasonicated one (raw sample).
- I am confused if the reference no. 32 can be used here as a reference. This seems to be the same unpublished manuscript.

Review form: Reviewer 2

Is the manuscript scientifically sound in its present form?

Yes

Are the interpretations and conclusions justified by the results?

Yes

Is the language acceptable?

No

Do you have any ethical concerns with this paper?

No

Have you any concerns about statistical analyses in this paper?

No

Recommendation?

Major revision is needed (please make suggestions in comments)

Comments to the Author(s)

Dear Authors,

I have reviewed your manuscript in detail.

There are several unclear points and many language errors in the paper which should be answered and corrected before publication.

Therefore, in my opinion, the paper needs major revisions.

Please find my review in the attached file (Appendix A).

Kind regards.

Decision letter (RSOS-191160.R0)

28-Aug-2019

Dear Dr Kang,

The editors assigned to your paper ("Intensifying the cleaning of flake graphite through ultrasonication") have now received comments from reviewers. We would like you to revise your paper in accordance with the referee and Associate Editor suggestions which can be found below (not including confidential reports to the Editor). Please note this decision does not guarantee eventual acceptance.

Please submit a copy of your revised paper before 20-Sep-2019. Please note that the revision deadline will expire at 00.00am on this date. If we do not hear from you within this time then it will be assumed that the paper has been withdrawn. In exceptional circumstances, extensions may be possible if agreed with the Editorial Office in advance. We do not allow multiple rounds of revision so we urge you to make every effort to fully address all of the comments at this stage. If deemed necessary by the Editors, your manuscript will be sent back to one or more of the original reviewers for assessment. If the original reviewers are not available, we may invite new reviewers.

When submitting your revised manuscript, you must respond to the comments made by the referees and upload a file "Response to Referees" in "Section 6 - File Upload". Please use this to document how you have responded to the comments, and the adjustments you have made. In

order to expedite the processing of the revised manuscript, please be as specific as possible in your response.

- Data accessibility

If you wish to submit your supporting data or code to Dryad (<http://datadryad.org/>), or modify your current submission to dryad, please use the following link:
<http://datadryad.org/submit?journalID=RSOS&manu=RSOS-191160>

- Competing interests

- Authors' contributions

- Acknowledgements

- Funding statement

on behalf of Dr Maria Charalambides (Associate Editor) and R. Kerry Rowe (Subject Editor)
openscience@royalsociety.org

Comments to Author:

Reviewers' Comments to Author:

Reviewer: 1

Comments to the Author(s)

Minor revision is recommended. This manuscript may be accepted after fulfilling the following comments:

- Written English of the manuscript needs to be improved for better clarity.
- Authors may add some lines or texts to the revised manuscript precisely indicating the importance of this research, its novelty and industrial relevance.
- Include these papers in the introduction section to support your work strongly:
DOI: <https://doi.org/10.1016/j.ultsonch.2019.04.033>
DOI: <https://doi.org/10.1016/j.ultsonch.2018.08.016>
DOI: <https://doi.org/10.1021/acs.energyfuels.9b01543>
- Include few more literatures particularly on the work published on flake graphite and discuss how the published work is different than your work. You can use this paper to have some preliminary idea:
DOI: <https://doi.org/10.1016/j.ultsonch.2019.04.033>
- Comparison between conventional and ultrasonicated flotation may be provided using recovery, yield and fixed carbon in the abstract.
- In section 2.2, authors may provide reference against the “orthogonal experiments” for better clarity of the optimization process being used.
- Authors have mentioned that, at higher power (1600 W), yield and recovery were found to be higher than those of 800 and 1200 W. Now the major concern here is its economic aspects since the higher power will consume more energy. Kindly justify this.
- What are the values of yield, recovery and fixed carbon content of conventional flotation? Please compare them for more clarity in section 3.1.
- In section 3.4, the elemental compositions of each feed and products varies differently. For instance, “Al and Fe” in the concentrate of conventional flotation is higher than its original sample. Similar trends can also be seen in concentrate of ultrasonicated flotation. Kindly comment on this.
- Contact angle on ultrasonicated graphite concentrate may be compared with those without ultrasonicated one (raw sample).

- I am confused if the reference no. 32 can be used here as a reference. This seems to be the same unpublished manuscript.

Reviewer: 2

Comments to the Author(s)

Dear Authors,

I have reviewed your manuscript in detail.

There are several unclear points and many language errors in the paper which should be answered and corrected before publication.

Therefore, in my opinion, the paper needs major revisions.

Please find my review in the attached file.

Kind regards.

Author's Response to Decision Letter for (RSOS-191160.R0)

See Appendix B.

RSOS-191160.R1 (Revision)

Review form: Reviewer 1

Is the manuscript scientifically sound in its present form?

Yes

Are the interpretations and conclusions justified by the results?

Yes

Is the language acceptable?

Yes

Do you have any ethical concerns with this paper?

No

Have you any concerns about statistical analyses in this paper?

No

Recommendation?

Accept as is

Comments to the Author(s)

The manuscript is well-revised and can be accepted. All the best!!!

Review form: Reviewer 2

Is the manuscript scientifically sound in its present form?

Yes

Are the interpretations and conclusions justified by the results?

Yes

Is the language acceptable?

Yes

Do you have any ethical concerns with this paper?

No

Have you any concerns about statistical analyses in this paper?

No

Recommendation?

Accept as is

Comments to the Author(s)

Dear Authors,

I have reviewed your responses and the revised manuscript in detail. You have responded the Reviewers' comments/questions and carried out the required revisions properly.

In my opinion, the manuscript can be accepted. However, there are still several minor and syntax errors are presented in the manuscript which I mentioned in the attached pdf file (Appendix C). Please check them and revise before publication.

Kind regards.

Decision letter (RSOS-191160.R1)

06-Nov-2019

Dear Dr Kang:

On behalf of the Editors, I am pleased to inform you that your Manuscript RSOS-191160.R1 entitled "Enhancement of flaky graphite cleaning by ultrasonic treatment" has been accepted for publication in Royal Society Open Science subject to minor revision in accordance with the referee suggestions. Please find the referees' comments at the end of this email.

The reviewers and Subject Editor have recommended publication, but also suggest some minor

revisions to your manuscript. Therefore, I invite you to respond to the comments and revise your manuscript.

Among the revisions recommended are assessing the language of the manuscript - please seek the advice of a service such as these listed at <https://royalsociety.org/journals/authors/language-polishing/>.

- Ethics statement

- Data accessibility

If you wish to submit your supporting data or code to Dryad (<http://datadryad.org/>), or modify your current submission to dryad, please use the following link:
<http://datadryad.org/submit?journalID=RSOS&manu=RSOS-191160.R1>

- Competing interests

- Authors' contributions

- Acknowledgements

- Funding statement

Because the schedule for publication is very tight, it is a condition of publication that you submit the revised version of your manuscript before 15-Nov-2019. Please note that the revision deadline will expire at 00.00am on this date. If you do not think you will be able to meet this date please let me know immediately.

on behalf of Dr Maria Charalambides (Associate Editor) and R. Kerry Rowe (Subject Editor)
openscience@royalsociety.org

Associate Editor Comments to Author (Dr Maria Charalambides):

Please act on the comment below by the Reviewer:

"However, there are still several minor and syntax errors are presented in the manuscript which I mentioned in the attached pdf file. Please check them and revise before publication."

Reviewer comments to Author:

Reviewer: 2

Comments to the Author(s)

Dear Authors,

I have reviewed your responses and the revised manuscript in detail. You have responded the Reviewers' comments/questions and carried out the required revisions properly.

In my opinion, the manuscript can be accepted. However, there are still several minor and syntax errors are presented in the manuscript which I mentioned in the attached pdf file. Please check them and revise before publication.

Kind regards.

Reviewer: 1

Comments to the Author(s)

The manuscript is well-revised and can be accepted. All the best!!!

Author's Response to Decision Letter for (RSOS-191160.R1)

See Appendix D.

Decision letter (RSOS-191160.R2)

27-Nov-2019

Dear Dr Kang,

It is a pleasure to accept your manuscript entitled "Enhancement of flaky graphite cleaning by ultrasonic treatment" in its current form for publication in Royal Society Open Science. The comments of the reviewer(s) who reviewed your manuscript are included at the foot of this letter.

on behalf of Dr Maria Charalambides (Associate Editor) and R. Kerry Rowe (Subject Editor)
openscience@royalsociety.org

Appendix A**ROYAL SOCIETY
OPEN SCIENCE****Intensifying the cleaning of flake graphite through
ultrasonication**

Journal:	Royal Society Open Science
Manuscript ID	RSOS-191160
Article Type:	Research
Date Submitted by the Author:	16-Jul-2019
Complete List of Authors:	Kang, Wenze; Heilongjiang University of Science and Technology, Institute of mining engineering Li, Huijian; Heilongjiang University of Science and Technology, Institute of mining engineering
Subject:	Materials science < ENGINEERING AND TECHNOLOGY
Keywords:	Ultrasound, Flotation, Surface Properties
Subject Category:	Engineering
Note: The following files were submitted by the author for peer review, but cannot be converted to PDF. You must view these files (e.g. movies) online.	
figure power point.rar table power point.rar	

Author-supplied statements

Relevant information will appear here if provided.

Ethics

Does your article include research that required ethical approval or permits?:

This article does not present research with ethical considerations

Statement (if applicable):

CUST_IF_YES_ETHICS :No data available.

Data

It is a condition of publication that data, code and materials supporting your paper are made publicly available. Does your paper present new data?:

My paper has no data

Statement (if applicable):

CUST_IF_YES_DATA :No data available.

Conflict of interest

I/We declare we have no competing interests

Statement (if applicable):

CUST_STATE_CONFLICT :No data available.

Authors' contributions

This paper has multiple authors and our individual contributions were as below

Statement (if applicable):

Kang Wenze and Li Huijian contributed to the design and execution of the study. Kang Wenze mainly contributed to the scheme design, experimental data analysis and the drafting and revision of papers. Li Huijian mainly contributed to the operation of experiments and data collation.

Intensifying the cleaning of flake graphite through ultrasonication

WenzeKang*, Huijian Li

Institute of mining engineering, Heilongjiang University of Science and Technology, Harbin 150022, China

Abstract: To accelerate the cleaning of graphite, experiments were carried out to examine intensification of the cleaning of graphite by using ultrasonic waves. On this basis, the influences of ultrasonic treatment on changes of particle sizes, morphologies, wettability, content of surface elements of flake graphite, and on the flotation effect of flake graphite were investigated. The results showed that ultrasonic treatment not only changed the size of flake graphite but also eliminated impurities on the graphite surface. Additionally, ultrasonic treatment improved the hydrophobicity of graphite. Compared with conventional flotation, flotation after ultrasonic treatment can remove not only silicate impurities but also most other metal impurities. Moreover, the yield, carbon content, and recovery of concentrates obtained through flotation based on ultrasonic treatment were significantly larger than those processed through conventional flotation. On the basis of carrying out ultrasonic treatment for 4 min, the carbon content in concentrates had reached 95% after only one flotation process, which satisfied industrial requirements imposed upon the cleaning of graphite. Intensifying the cleaning of graphite by ultrasonication can accelerate the cleaning process of graphite.

Keywords: Ultrasound; Flotation; Surface Properties

1 Introduction

Flotation is one of main methods for separating fine-grained minerals. Owing to there being different types of fine-grained minerals, the process of flotation also varies. Generally, it can satisfy the requirement of separating fine coal only through one flotation cycle, however, for flotation of graphite, one flotation operation cannot meet industrial requirements. Owing to the low grade of raw graphite (generally, in the range of 5~15%) and high requirement imposed upon the final grade of concentrates (a carbon content of greater than 95%), the required purity of the concentrate cannot be reached through one flotation operation. The requirement cannot be met unless multiple grinding and flotation processes are carried out. For example, in a certain graphite concentrator in Luobei County, Heilongjiang Province, China, the grade of raw graphite was 10.38%, and the carbon content in graphite cannot reach 95% before being subjected to rough grinding, grading, roughing, then nine ore-grinding cycles and ten cleaning operations. In the cleaning process, on condition that the carbon content in graphite rose to 90%, it was necessary to conduct one ore-grinding step and one cleaning step each time a 1% increase in carbon content in graphite was needed: this entailed a long flotation process and high energy consumption. Therefore, it is necessary to intensify the cleaning of graphite by selecting a technological method with which to shorten the flotation process for graphite.

In recent years, research into application of ultrasonic treatment in intensifying mineral flotation has gradually increased. Many scholars have explored the flotation process of oxidized coal and coal with a high ash content assisted with ultrasonic waves. The research showed that, at the same dosage, a higher yield of clean coal and a lower ash content in the clean coal can be attained by conducting flotation based on ultrasonic treatment [1-8]. Scholars explored the influence of ultrasonic waves on the flotation of high-sulphur coal to reveal that ultrasonic waves can promote the dissociation between coal and pyrite and increase the difference of floatability between coal and pyrite. The flotation based on ultrasonic treatment for high-sulphur coal showed remarkable de-ashing and de-sulphurising effects [9-17]. Scholars also explored the influence of ultrasonic treatment on the flotation of borocalcite-clay, magnesite-quartz, lead sulphide-xanthate, fluorite-quartz, barite-copper pyrites, and borate-josephinite. The research indicated that ultrasonic treatment can selectively change the surface properties of targeted mineral particles, enhance the selectivity of flotation, and improve the flotation efficiency [18-22]. Letmaheet *al.* [23] investigated the flotation process of graphite and reported the influence of ultrasonic treatment on the flotation effect of graphite. After conducting ultrasonic treatment, reagents were more uniformly distributed in suspension so as to increase the effect thereof, thus increasing the purity of concentrates and decreasing the ash content of concentrates.

Although there have been some remarkable achievements made when intensifying the flotation of minerals through ultrasonic waves, the research on intensifying the cleaning of graphite by using ultrasonic waves is sparse: to enrich the research on intensifying the cleaning of graphite through ultrasonic waves, a method for shortening this cleaning process was explored. Herein, the influences of ultrasonic treatment on changes of particle sizes, morphologies, wettability, and surface element contents of flake graphite and on the flotation effect were investigated.

2 Test materials and methods

2.1 Samples

The test samples taken from a certain graphite concentrator in Luobei County, Heilongjiang Province, China and subjected to nine ore-grinding, and 10 cleaning, stages. At the sixth concentrate slot for cleaning, the samples were collected in a sampling bucket (1 L volume) at 15 min intervals. After continuous sampling, the samples were collected and blended, dewatered, dried, and then bagged, with the total mass there of being 30 kg. The average carbon content of the samples was 90.56%.

2.2 Test on flotation and ultrasonic treatment

AnXFD-1.0 flotation machine was applied, with the volume of the flotation cell of 1,000 mL. Kerosene was taken as collector while 2-octanol was used as a frother. By selecting three main influencing factors (the pulp density and the dosages of kerosene and 2-octanol), the optimal parameters (involving pulp density and dosages of kerosene and 2-octanol of $40\text{g}\cdot\text{L}^{-1}$, $300\text{g}\cdot\text{t}^{-1}$ and $125\text{g}\cdot\text{t}^{-1}$, respectively) for flotation were determined through orthogonal experiment.

In the ultrasonic experiment, a JAC-5500 ultrasonic generator was employed, working at the frequency of 25 kHz and output power of up to 2 kW. The samples were subjected to ultrasonic treatment at 800, 1200, or 1600 W, for 2, 3, 4, 5, and 6 min, respectively.

In the test, 40 g graphite was placed into a beaker, to which 500 mL water was added. The mixture was stirred and then ultrasonically treated. The ultrasonicated pulp was removed and cooled to room temperature. Subsequently, the cooled pulp was put into the flotation machine and stirred for 2 min, into which kerosene was added and the mixture stirred for 1 min. This was followed by the addition of 2-octanol into the mixture and stirring for 30 s and flotation for 2 min. Next, the froth and tailings of concentrates were collected, filtered, and dried and the dried samples were weighed to calculate the yield of the products. By utilising high-temperature burning, the ash content and volatile matter in graphite were measured according to Chinese Standard GB T 3521-2008.

The carbon content of the flotation products is given by:

$$\beta = 100 - A_d - V_d \quad (1)$$

where, β , A_d , and V_d denote the carbon content in products (%), the ash content (%), and volatile matter (%), respectively.

The recovery of products is given by:

$$\varepsilon = (\beta \times \gamma_j / \alpha) \times 100\% \quad (2)$$

where, ε , α , γ_j , and β represent recovery (%), carbon content in original samples (%), yield of products (%), and carbon content in products (%), separately.

3 Test results and discussion

3.1 The influence of ultrasonic treatment under different powers on the flotation effect of flake graphite

According to Section 2.2, flotation based on ultrasonic treatment under three powers (800, 1200, and 1600 W) was conducted to compare the yields, carbon contents, and recoveries of concentrates under different powers (Figure 1). Only conventional flotation was performed in the absence of ultrasonic treatment.

As shown in Figures 1(a), (b), and (c), the yield of concentrates treated by using ultrasonic waves was greater than that from specimens processed through conventional flotation. The carbon content in concentrates obtained through flotation after ultrasonic treatment for 2 min was 2% higher than that attained using conventional flotation. On condition that the ultrasonic treatment was conducted for longer than 4 min, the carbon content in concentrates obtained through flotation exceeded 95%. Moreover, the recovery of ultrasonically treated concentrates was also much greater than that attained based on conventional flotation. At 800, 1200, and 1600 W, with the growth of ultrasonic power, the carbon content of concentrates did not change significantly while the yield and recovery of the concentrates both increased. There was the optimal flotation effect when the ultrasonic power was 1600 W. Through the aforementioned analysis, the carbon content of concentrates reached 95% after only one flotation step after ultrasonic treatment. This test verified that the introduction of ultrasonic waves can simplify the cleaning process of graphite. Owing to the optimal flotation effect being found at an ultrasonic power of 1600 W, this setting was applied in subsequent ultrasonication trials.

(a) Yield of concentrates of graphite flotation with different ultrasound power

(b) Carbon content of concentrates of graphite flotation with different ultrasound power

(c) Recovery of concentrates of graphite flotation with different ultrasound power

Figure 1 Effect of graphite flotation with different ultrasonic power

3.2 The influences of ultrasonic treatment on particle size and morphologies of flake graphite

3.2.1 The influence of ultrasonic treatment on morphologies of flake graphite

Some 40 g of graphite was uniformly mixed with 500 mL of water, which was subjected to ultrasonic treatment for 5 min and then filtered and dried. Afterwards, the graphite with a same mass was taken and uniformly mixed with 500 mL of water and the mixture was filtered and dried. By using an MX2600 scanning electron microscope (SEM), the surface morphologies of graphite, in ultrasonicated, and untreated states were separately detected (Figure 2).

(a) Surface morphology of graphite without ultrasonic treatment

(b) Surface morphology of graphite with ultrasonic treatment

Figure 2 Surface topography of graphite

Figures 2(a) and 2(b) separately present SEM images of untreated graphite and that treated by using ultrasonic waves for 5 min. Figure 2(a) shows that fine flaky bright components were found on the graphite surface, which were associated minerals and fine silt adsorbed onto the graphite surface. As shown in Figure 2(b), the fine bright components were significantly reduced, implying that the associated minerals or fine silt adsorbed onto the graphite surface were eliminated after ultrasonic treatment. By conducting ultrasonic treatment, the impurities on the graphite surface were cleaned off to expose a fresh surface of the graphite[24,25]. On the one hand, it revealed that the carbon content in graphite had been increased; on the other hand, it indicated that the hydrophobicity of graphite increased after ultrasonication.

3.2.2 The influence of ultrasonic treatment on particle size of flake graphite

During the test, 40 g of graphite was uniformly mixed with 500 mL of water and then the mixture was

ultrasonically treated for 5 min. Afterwards, wet screening was carried out. Graphite with the same mass, but untreated, was also wet screened (Table 1): for the particle-size grades of +0.25, 0.25-0.15 and 0.15-0.074 mm, the yields of graphite at different grades after ultrasonic treatment were lower than those without ultrasonic treatment. This indicated that ultrasonic treatment showed a significant breaking effect on graphite with larger particles[26]. For grades 0.074~0.045 and -0.045 mm, the yields of graphite at various grades undergoing ultrasonic treatment were higher than those without. It was possible that the graphite at the grades of +0.25, 0.25-0.15 and 0.15-0.074 mm was broken to that at grades of 0.074~0.045 and -0.045 mm, thus, the yields of graphite with grades of 0.074~0.045 and -0.045 mm increased. Ultrasonic waves can also break graphite at the grades of 0.074~0.045 and -0.045 mm, however, owing to the graphite at these grades being too small and some broken fine particles of graphite at the other grades interfered with the results, this failed to reflect the breaking effect of ultrasonic waves on graphite at these two grades. The carbon contents in graphite at five grades all increased after ultrasonic treatment and ultrasonic treatment can eliminate impurities on graphite surface. In the filtering process, fine impurities were adsorbed onto filter paper, thus resulting in the increase of carbon content in graphite. Based on the yields and carbon contents with different particle sizes in graphite being subjected to ultrasonic treatment, it can be seen that ultrasonic treatment showed a certain breaking and cleaning effect on graphite.

Table 1 Test results of graphite screening (%)

Particle-size grade/mm	Unultrasonic treatment		Ultrasonic treatment		Grade change rate	
	Yield	Carbon content	Yield	Carbon content	Yield	Carbon content
+0.25	1.44	94.76	0.77	95.60	-46.53	0.89
0.25-0.15	9.71	94.15	3.59	95.27	-63.03	1.19
0.15-0.074	37.32	92.93	20.19	94.35	-45.90	1.54
0.074-0.045	8.34	91.55	8.59	92.38	3.00	0.91
-0.045	43.19	89.14	66.85	90.43	54.78	1.45

3.3 The influence of ultrasonic treatment on the wettability of flake graphite

3.3.1 The influence of ultrasonic treatment on the contact angle of flake graphite

In the test, based on direct method, the contact angle of graphite was measured using an XG-CAMA1 enhanced static contact goniometer. Within the die with a diameter of 40 mm, 2 g of graphite powders were put in each time and pressed into round cakes under a constant pressure of 10 MPa for 2 min. A smooth zone was selected from the cakes. By using a microlitre syringe, the test solution (distilled water) was placed onto the graphite surface to form hemispherical drops, each with a diameter of about 1 mm and the drops photographed. According to the height of the drops and the radius of the bottom surface, the contact angle can be calculated. Three points were selected from each circular graphite flake to measure the corresponding contact angle (Table 2).

Table 2 Contact angle measured after graphite pressed into cakes

Ultrasound treatment time/min	0	2	3	4	5	6
Contact angle /°	86.37	87.15	88.11	89.34	89.61	89.56

It can be seen from Table 2 that with increasing ultrasonic treatment time, the contact angle between graphite and water gradually increased. After longer than 5 min, the contact angle did not change to any further significant extent. Ultrasonic treatment caused an increase in contact angle between graphite and water, which implied that the treatment strengthened the hydrophobicity of graphite. The reason was likely to have been such that, under the cavitation generated due to the propagation of ultrasonic waves, various impurities, such as clay particles, adsorbed on the graphite surface were eliminated, thus exposing a fresh surface beneath, thus increasing the hydrophobicity[27-29].

3.3.2 The influence of ultrasonic treatment on the wetting heat of flake graphite

Generally, water is used as the wetting liquid to measure the wetting heat of minerals. The flake graphite

shows a low density and a strong hydrophobicity and is flaky so it cannot be completely wetted by water. Therefore, the wetting heat cannot be accurately measured by using water as the wetting liquid. Through testing, it can be found that kerosene can wet flake graphite and thus kerosene was selected as the wetting liquid herein.

By using a SetaramC80D microcalorimeter, the wetting heat of graphite was measured. In each test, 0.5 g of graphite was weighed and 2 mL of kerosene (the wetting liquid) was taken. The samples were kept at the initial temperature of 26°C for 2 h during the test and the data on wetting heat were collected over a 3-h period.

The prepared graphite samples were ultrasonically treated for 5 min. Ultrasonic treatment changed the particle size of graphite, therefore, to reflect changes in the wetting heat in graphite without undergoing ultrasonic treatment and in ultrasonically treated graphite, wet screening was separately carried out on the two types of graphite. As a result, the graphite was screened into four particle-size grades (0.25~0.15, 0.15~0.74, 0.074~0.045, and -0.045 mm). By comparing the wetting heat in graphite without, and with, ultrasonic treatment at the same grade (Table 3), the influence of ultrasonic treatment on the wettability of graphite can be reflected [30].

Table 3 The wetting heat of graphite and kerosene with different particle size ($\text{J} \cdot \text{g}^{-1}$)

Particle-size grades/mm	Graphite	Ultrasound-treated graphite
0.25-0.15	0.071	0.148
0.15-0.74	0.083	0.300
0.074-0.045	0.307	0.326
-0.045	0.489	0.561

As shown in Table 3, compared with graphite without undergoing ultrasonic treatment, the ultrasonically treated graphite at each grade always exhibited an increased wetting heat with kerosene. This indicated that the interaction between ultrasonically treated graphite and kerosene was strengthened. According to the “like dissolves like” principle, the more similar the properties of a mineral and the wetting liquid, the stronger the interaction between the two parameters and the greater the wetting heat [31]. Kerosene is hydrophobic, so the wetting heat between kerosene and ultrasonically treated graphite was larger than that between kerosene and untreated graphite. This indicated that the surface properties of ultrasonically treated graphite were more similar to those of hydrophobic kerosene. By analysing the changes in wetting heat of ultrasonically treated graphite and untreated graphite, it can be inferred that the hydrophobicity of graphite increased after ultrasonication. This conformed to the fact that, after being subjected to ultrasonic treatment, the contact angle between the graphite and water increased and hydrophobicity thereof increased.

3.4 The influence of ultrasonic treatment on the element content of flake graphite surface

Under X-ray irradiation, each matter emits a specific X-ray signal. A fluorescence spectrometer can capture these characteristic emissions to analyse wavelengths and determine the corresponding elements present. In the test, the contents of mineral elements were measured using an S4EXPLORER spectrometer. The original samples, concentrates obtained through conventional flotation, ultrasonically treated samples, and concentrates acquired through flotation based on ultrasonic treatment were used here. The measured results are shown in Table 4.

Table 4 The measured results of main elements in graphite%

Element	Original sample/%	Concentrate by conventional flotation/%	Ultrasonically treated samples/%	Concentrate by ultrasonic flotation/%
Al	6.398	6.617	8.543	5.5
Si	18.109	15.652	22.06	13.895
P	0.423	0.5	1.057	1.743
S	7.374	5.262	5.758	6.716
K	7.541	8.48	12.497	7.424
Ca	9.085	5.33	6.9	5.23
Fe	41.749	47.051	31.51	46.327
Cl	1.639	2.003	2.313	2.639
Ag	0.781	0.942	0.541	1.473

It can be seen (Table 4) that, compared with the original samples, the elemental Si, S, and Ca contents in
concentrates obtained by using conventional flotation all decreased. This indicated that conventional flotation can
decrease the amount of impurities and increase the carbon content in concentrates to some extent, however, **except**
**for Ca whose elemental content decreased**, the amounts of all other metal elements assayed increased. This implied

[revised manuscript text omitted]

32. Kang WZ, Li HJ. 2019 Intensifying the cleaning of flake graphite through ultrasonication. *Dryad Digital Repository*. <https://datadryad.org/review?doi=doi:10.5061/dryad.98521v6>.

(a) Yield of concentrates of graphite flotation with different ultrasound power

(b) Carbon content of concentrates of graphite flotation with different ultrasound power

(c) Recovery of concentrates of graphite flotation with different ultrasound power

Figure 1 Effect of graphite flotation with different ultrasonic power

(a) Surface morphology of graphite without ultrasonic treatment

(b) Surface morphology of graphite with ultrasonic treatment

Figure 2 Surface topography of graphite

Table 1 Test results of graphite screening (%)

Particle-size grade/mm	Unultrasonic treatment		Ultrasonic treatment		Grade change rate	
	Yield	Carbon content	Yield	Carbon content	Yield	Carbon content
+0.25	1.44	94.76	0.77	95.60	-46.53	0.89
0.25-0.15	9.71	94.15	3.59	95.27	-63.03	1.19
0.15-0.074	37.32	92.93	20.19	94.35	-45.90	1.54
0.074-0.045	8.34	91.55	8.59	92.38	3.00	0.91
-0.045	43.19	89.14	66.85	90.43	54.78	1.45

Table 2 Contact angle measured after graphite pressed into cakes

Ultrasound treatment time /min	0	2	3	4	5	6
Contact angle /°	86.37	87.15	88.11	89.34	89.61	89.56

Table 3 The wetting heat of graphite and kerosene with different particle size ($\text{J} \cdot \text{g}^{-1}$)

Particle-size grades/mm	Graphite	Ultrasound-treated graphite
0.25-0.15	0.071	0.148
0.15-0.74	0.083	0.300
0.074-0.045	0.307	0.326
-0.045	0.489	0.561

Table 4 The measured results of main elements in graphite%

Element	Original sample /%	Concentrate by conventional flotation /%	Ultrasonically treated samples /%	Concentrate by ultrasonic flotation /%
Al	6.398	6.617	8.543	5.5
Si	18.109	15.652	22.06	13.895
P	0.423	0.5	1.057	1.743
S	7.374	5.262	5.758	6.716
K	7.541	8.48	12.497	7.424
Ca	9.085	5.33	6.9	5.23
Fe	41.749	47.051	31.51	46.327
Cl	1.639	2.003	2.313	2.639
Ag	0.781	0.942	0.541	1.473

Appendix B

Reply for the comments of reviewer 1

General comments of reviewer 1:

Slight modifications in text and figure.9 recommended as indicated in paper.

Minor revision is recommended. This manuscript may be accepted after fulfilling the following comments. Written English of the manuscript needs to be improved for better clarity.

According to the comments of review 1, we answered the questions and made modifications as following:

1. Authors may add some lines or texts to the revised manuscript precisely indicating the importance of this research, its novelty and industrial relevance.

Answer:

The emphasis of this paper is on the enhancement of graphite flotation by ultrasound. The aim is to simplify the graphite cleaning process and promote the efficient development of graphite industry. In this paper, the flow-sheet of conventional flotation and the flow- sheet of ultrasonic treatment flotation are added to show that intensifying graphite flotation by ultrasonic treatment can shorten the graphite cleaning process. Technological process flow-sheet of the graphite concentrator in Heilongjiang, China is shown in figure 1. Technological graphite flotation processing flow-sheet using ultrasonic treatment is shown in the figure 3.

Figure.1 Technological processing flow-sheet of the Luobei graphite concentrator

Figure.3 Technological graphite flotation processing flow-sheet using ultrasonic treatment

2. Include these papers in the introduction section to support your work strongly:

DOI: <https://doi.org/10.1016/j.ultsonch.2019.04.033>—

DOI: <https://doi.org/10.1016/j.ultsonch.2018.08.016>—

DOI: <https://doi.org/10.1021/acs.energyfuels.9b01543>—

Answer:

These three papers have been added to the introduction as references 21, 22 and 29.

3. Include few more literatures particularly on the work published on flake graphite and discuss how the published work is different than your work. You can use this paper to have some preliminary idea: DOI: <https://doi.org/10.1016/j.ultsonch.2019.04.033>

Answer:

The paper recommended by the reviewer was cited in the introduction and listed in the

references. The following sentences were added to the introduction. Santosh DebBarma et al.[29] investigated the feasibility of low-frequency ultrasound in enhancing the flotability of flaky graphite from low-grade graphite ore. Their results showed that the yield, fixed carbon content and percentage recovery of the flotation concentrate products increased significantly under ultrasonic-assisted flotation.

The emphasis of my study is on the enhancement of graphite flotation by ultrasound. The aim is to simplify the graphite cleaning process and promote the efficient development of graphite industry.

4. Comparison between conventional and ultrasonicated flotation may be provided using recovery, yield and fixed carbon in the abstract.

Answer:

In the abstract section, the following paragraph was added according to the reviewer's suggestion.

“The yield, carbon content and recovery of flotation concentrate were 91.46%, 95.17% and 96.12% after ultrasonic treatment for 4 minutes with ultrasound power 1600W, which were 5.83%, 2.86% and 8.84% higher than conventional flotation, respectively.”

5. In section 2.2, authors may provide reference against the “orthogonal experiments” for better clarity of the optimization process being used.

Answer:

We did the orthogonal experiment. The process of the experiment was as follows. It was limited to the length of the article and was not put in the text.

On the basis of single factor test, the main factors such as pulp concentration (A), dosage of kerosene (B) and dosage of secondary octyl alcohol (C) were selected. Design orthogonal experiments to determine the optimum parameters. The orthogonal experimental design is shown in table 1 and the flotation experimental results are shown in table 2.

On the basis of single factor test, the main factors such as pulp concentration (A), dosage of kerosene (B) and dosage of secondary octyl alcohol (C) were selected. Design orthogonal experiments to determine the optimum parameters. The orthogonal experimental design is shown in table 1 and the flotation experimental results are shown in table 2.

Table 1 Orthogonal experiment design

	A	B	C
level	/g·L ⁻¹	/g·t ⁻¹	/g·t ⁻¹
1	40	150	75
2	50	200	100
3	60	250	125

4

70

300

150

Table 2 Flotation experimental results

number	A /g·L ⁻¹	B / g·t ⁻¹	C / g·t ⁻¹	yield /%	carbon content /%	recovery /%
A1B1C1	40	150	75	77.24	91.75	79.45
A1B2C2	40	200	100	78.31	92.76	81.22
A1B3C3	40	250	125	85.09	92.27	88.06
A1B4C4	40	300	150	86.41	92.08	89.05
A2B1C2	50	150	100	77.79	92.55	80.51
A2B2C1	50	200	75	78.36	92.22	81.01
A2B3C4	50	250	150	77.24	91.58	79.00
A2B4C3	50	300	125	78.67	92.10	81.05
A3B1C3	60	150	125	81.23	92.66	83.93
A3B2C4	60	200	150	75.80	92.39	78.28
A3B3C1	60	250	75	75.93	92.13	78.47
A3B4C2	60	300	100	75.38	92.37	77.69
A4B1C4	70	150	150	79.12	92.53	81.57
A4B2C3	70	200	125	77.19	92.19	79.62
A4B3C2	70	250	100	73.27	91.97	75.59
A4B4C1	70	300	75	79.72	92.27	82.43

According to the flotation experiment results in table 2, the average values of concentrate yield, carbon content and recovery of products with different factors and levels are calculated. The results are shown in table 3, Table 4 and Table 5.

Table 3 Calculation results of average yield of concentrate

Factor	1	2	3	4
A	81.76	78.01	77.08	77.33
B	78.84	77.41	77.88	80.04
C	77.81	76.19	80.55	79.64

Table 4 Calculation results of average carbon content of concentrate

Factor	1	2	3	4
A	92.21	92.11	92.39	92.24

B	92.37	92.39	91.99	92.21
C	92.09	92.41	92.30	92.15

Table5 Calculation results of average recovery of concentrate

Factor	1	2	3 ₃	4
A	84.44	80.39	79.59	79.80
B	81.37	80.03	80.28	82.56
C	80.34	78.75	83.17	81.98

Analyzing the data in table 3, 4 and 5, considering comprehensively, the optimal combination is A1B4C3, i.e. 40g. L⁻¹ of the pulp concentration, 300 g. t⁻¹ of the kerosene consumption and 125 g. t⁻¹ of the secondary octyl alcohol consumption are the optimal combination.

6. Authors have mentioned that, at higher power (1600 W), yield and recovery were found to be higher than those of 800 and 1200 W. Now the major concern here is its economic aspects since the higher power will consume more energy. Kindly justify this.

Answer:

For example, when the ultrasound treatment time is 4 min, the concentrate yield of 800W, 1200W and 1600W is 86.56%, 89.12% and 91.46% respectively.

200 g of original sample can be processed for 4 min in JAC-5500 Ultrasound Generator every time. It's 0.5 yuan RMB per kilowatt hour. The electricity charges for 800W, 1200W and 1600W of a ton of original sample are 133, 200 and 267 yuan RMB respectively. The electricity charges for 1600W ultrasound are 134 and 67 yuan RMB more than that of 800W and 1200W.

The market price of graphite products with 95% carbon content is 4500 yuan RMB per ton. Because of the high concentrate yield obtained by 1600W ultrasound, the income of treating 1 ton of original 1600W ultrasound is 245 yuan RMB and 117 yuan RMB more than that of 800W and 1200W respectively.

The income of a ton of original sample with 1600W ultrasonic treatment is 111 yuan RMB and 50 yuan RMB more than that of 800W and 1200W respectively. Consequently, the economic benefits of 1600W ultrasound are better.

7. What are the values of yield, recovery and fixed carbon content of conventional flotation? Please compare them for more clarity in section 3.1.

Answer:

In section 3.1, the second paragraph, the following paragraph was added.

Points of the ultrasonic treatment time 0 min in the figures 2(a), (b), and (c) represent the products parameters obtained by conventional flotation. The yield, carbon content and recovery of the concentrate products of conventional flotation are 85.63%, 92.31% and 87.28%, respectively.

8. In section 3.4, the elemental compositions of each feed and products varies differently. For instance, “Al and Fe” in the concentrate of conventional flotation is higher than its original sample. Similar trends can also be seen in concentrate of ultrasonicated flotation. Kindly comment on this.

Answer:

Fluorescence spectrometry is used to determine the relative content of elements in samples. The content of elements in each sample is calculated by 100%. After ultrasonic treatment, some minerals may be separated from the sample, resulting in the reduction of some elements in the sample. As the content of some elements decreases, the content of other elements is increased relatively. For example, the content of Si, S and Ca after conventional flotation is lower, while the content of Al, K, Fe and Ag is higher than that in original samples.

9. Contact angle on ultrasonicated graphite concentrate may be compared with those without ultrasonicated one (raw sample).

Answer:

In this study, the contact angle of the ultrasonic treatment graphite concentrate was compared with those without ultrasonic treatment one (raw graphite). Table 2 shows the contact angle of the ultrasonic treatment graphite concentrate in different ultrasonic treatment time and those without ultrasonicated one. When the ultrasonic treatment time is 0 min, i.e., without ultrasonic treatment, the contact angle of graphite powder is 86.37°.

The sentence “Six graphite powder samples were prepared, one of which was not treated by ultrasound and the other five samples were treated by ultrasound for 2,3,4,5 and 6 min respectively.” was added to the first paragraph of section 3.3.1.

The sentence “In this study, the contact angle of the ultrasonic treatment graphite was compared with that of the raw graphite. Table 2 shows when the ultrasonic treatment time is 0 min, i.e., without ultrasonic treatment, the contact angle of the raw graphite pellet is 86.37°.” was added to the second paragraph of section 3.3.1.

10. I am confused if the reference no. 32 can be used here as a reference. This seems to be the same unpublished manuscript.

Answer:

According to open data policies of Royal Society Open Science, it is a condition of submission and publication that all supporting data for published articles are made freely available either as supplementary information or preferably via an appropriate repository.

This link will only be active while my manuscript is under review. It will allow editors and reviewers to access my data while my manuscript is under review.

I may include the temporary review link in my manuscript now for reviewers, but that it will no longer work when my data are publicly archived. If/when my manuscript is accepted for

publication, and my data are publicly archived, the link to access my data will change to the permanent DOI.

11. Written English of the manuscript needs to be improved for better clarity.

Answer:

We noticed some spaces between words are lost when word files are converted into PDF files and corrected them.

Written English of the manuscript has been revised for better clarity especially the abstract section. The abstract revised as follows:

Abstract: In this study, the aim is to simplify the graphite cleaning processing. In order to achieve flotation for graphite effectively, ultrasonic treatment was utilized as a pre-treatment technique. Flotation tests were conducted using for different ultrasound power and ultrasonic treatment time. The influences of ultrasonic treatment on particle sizes, morphologies, wettability, content of surface elements and on the flotation effect of flaky graphite were investigated. The results of ultrasonic treatment for graphite flotation were compared to the results of conventional flotation. The results showed that ultrasonic treatment not only changed the size of flaky graphite but also eliminated impurities on the graphite surface. Additionally, ultrasonic treatment improved the hydrophobicity of graphite. The effects of ultrasound were observed that it can remove not only silicate impurities but also most other metal impurities. The yield, carbon content and recovery of flotation concentrate were 91.46%, 95.17% and 96.12% after ultrasonic treatment for 4 minutes with ultrasound power 1600W, which were 5.83%, 2.86% and 8.84% higher than that of conventional flotation, respectively. The graphite after ultrasonic treatment was conducted only one times flotation, the carbon content in concentrates products had reached 95%. This study indicates that intensifying graphite flotation by ultrasonic treatment can shorten the graphite cleaning process.

Reply for the comments of reviewer 2

General comments of reviewer 2:

There are several unclear points and many language errors in the paper which should be answered and corrected before publication. Therefore, in my opinion, the paper needs major revisions.

According to the comments of review 2, we answered the questions and made modifications as following:

1. Page 2, line 4, ultrasonication There are many different terms for ultrasonic treatment in the paper(with/without/based on ultrasonic treatment or ultrasonic waves, ultrasonication, application of ultrasonic treatment, unultrasonic treatment) This has a significant negative effect on the understandability of the paper. Please chose one of them for "ultrasonic treatment" and one other for the "no ultrasonic treatment" and use them throughout the paper.

Answer:

According to the reviewer's suggestion, we have revised the English for ultrasound and non-ultrasound treatment throughout the paper.

2. Page 2, line 18, 4 min the power of ultrasound can be added in the abstract.

Answer:

In section abstract, the following paragraph was added according to the reviewer's suggestion.

The yield, carbon content and recovery of flotation concentrate were 91.46%, 95.17% and 96.12% after ultrasonic treatment for 4 minutes with ultrasound power 1600W, which were 5.83%, 2.86% and 8.84% higher than conventional flotation, respectively.

3. Page 2, line 1, Keywords: "Graphite" should be added to the keywords.

Answer:

The words "Graphite" was added to the keywords according to the reviewer comment.

4. Page 2, line 29, Requirements: citation required

Answer:

The following references were added in the first paragraph in the introduction section.

Reference 1. Chelgani SC, Rudolph M, Kratzsch R, Sandmann D, Gutzmer J. 2016 A review of graphite beneficiation techniques, *Miner. Process. Extr. Metall. Rev.* 37, 58–68.

Doi:10.1080/08827508.2015.1115992

Reference 2. Vasumathi N, Vijaya Kumar TV, Ratchambigai S, Subba Rao S, Bhaskar Raju G, 2015, Flotation studies on low grade graphite ore from eastern India. *Int. J. Min. Sci. Technol.* 25, 415–420 DOI: 10.3969/j.issn.2095-2686.2015.03.013

Reference 3.W. Peng, Y. Qiu, L. Zhang, J. Guan, S. Song, 2017Increasing the fine flaky graphite recovery in flotation via a combined multiple treatments technique of middlings, Minerals 7(11): 208.DOI: 10.3390/min7110208

5. Page 2, line 29, “in the range of 5~15%”, is it 5~15% C?

Answer:

Yes, it is. The sentence changed to “the range of carbon content is 5~15%”

6. Page 2, line 40 “application of ultrasonic treatment”, “application” and “treatment” are the words with almost the same meaning

Answer:

The word “application” was deleted.

7. Page 2, line 42, “The research”: which one?

Answer:

“The research” changed to “Existing studies”. It is not a single research, but it is several researches such as references 4-11.

8. Page 2, line 44, “Scholars” changed to “Some researchers investigated”.

9. Page 2, line 47, “Scholars” changed to “Some other researchers studied”.

10. Page 2, line 49, “The research”: which one?

Answer:

“The research” changed to “Their researches”. It is not a single research, but It is several researches such as references 23-27.

11. Page 2, line 53, “so as to increase the effect thereof,” incomprehensible sentence

Answer:

Original sentence “After conducting ultrasonic treatment, reagents were more uniformly distributed in suspension so as to increase the effect thereof, thus increasing the purity of concentrates and decreasing the ash content of concentrates.” was deleted and changed to “Ultrasonic treatment can improve the effectiveness of a reagent due to a more uniform distribution in the suspension.”

12. Page 2, line 56, “sparse”: There are many studies on the ultrasonic application on graphite.

Please specify the difference of your study from these studies.

Answer:

Previous researchers have made remarkable achievements in enhancing minerals flotation by ultrasound, but the effect of ultrasonic treatment on graphite cleaning process is studied less. To develop a method for shortening the graphite cleaning process, the influences of ultrasonic treatment on changes of particle sizes, morphologies, wettability, content of surface elements of flaky graphite, and on the flotation effect of flaky graphite were explored in this paper. The emphasis of this paper is on the enhancement of graphite flotation by ultrasound. The aim is to simplify the graphite cleaning process and promote the efficient development of graphite industry.

13. Page 2, line 58, “explored...” in this study?

Answer:

Yes, it is. So, the sentence changed to “...this cleaning process was explored in this study.”

14. Page 3, line 7 “taken” changed to “were taken”

15. Page 3, line 8, “cleaning”, “what is the method for cleaning the graphite in this concentrator?”

Answer:

For illustration the method for cleaning the graphite in this concentrator, the section 2.1 changed as following:

The test samples were taken from a graphite concentrator in Luobei, Heilongjiang Province, China. The beneficiation process of the concentrator consisted of crushing, coarse and rougher, nine stages regrinding and ten stages cleaning as shown in figure1. The flotation method is the method for cleaning the graphite in this concentrator.

16. Page 3, line 14, “ applied” changed to “employed”

17. Page 3, line 17, “determined” Why did not you give the results of these optimization experiments in the paper?

Answer:

We did the orthogonal experiment. The process of the experiment was as follows. It was limited to the length of the article and was not put in the text.

On the basis of single factor test, the main factors such as pulp concentration (A), dosage of kerosene (B) and dosage of secondary octyl alcohol (C) were selected. Design orthogonal experiments to determine the optimum parameters. The orthogonal experimental design is shown in Table 1 and the flotation experimental results are shown in Table 2.

On the basis of single factor test, the main factors such as pulp concentration (A), dosage of kerosene (B) and dosage of secondary octyl alcohol (C) were selected. Design orthogonal experiments to determine the optimum parameters. The orthogonal experimental design is shown in Table 1 and the flotation experimental results are shown in Table 2.

Table 1 Orthogonal experiment design

level	A	B	C
	/g·L-1	/ g·t-1	/ g·t-1
1	40	150	75
2	50	200	100
3	60	250	125
4	70	300	150

Table 2 Flotation experimental results

number	A	B	C	yield	carbon	recovery
	/g·L-1	/ g·t-1	/ g·t-1	/%	content	
					/%	/%
A1B1C1	40	150	75	77.24	91.75	79.45
A1B2C2	40	200	100	78.31	92.76	81.22
A1B3C3	40	250	125	85.09	92.27	88.06
A1B4C4	40	300	150	86.41	92.08	89.05
A2B1C2	50	150	100	77.79	92.55	80.51
A2B2C1	50	200	75	78.36	92.22	81.01
A2B3C4	50	250	150	77.24	91.58	79.00
A2B4C3	50	300	125	78.67	92.10	81.05
A3B1C3	60	150	125	81.23	92.66	83.93
A3B2C4	60	200	150	75.80	92.39	78.28
A3B3C1	60	250	75	75.93	92.13	78.47
A3B4C2	60	300	100	75.38	92.37	77.69
A4B1C4	70	150	150	79.12	92.53	81.57
A4B2C3	70	200	125	77.19	92.19	79.62
A4B3C2	70	250	100	73.27	91.97	75.59
A4B4C1	70	300	75	79.72	92.27	82.43

According to the flotation experiment results in Table 2, the average values of concentrate yield, carbon content and recovery of products with different factors and levels are calculated. The results are shown in Table 3, Table 4 and Table 5.

Table 3 Calculation results of average yield of concentrate

Factor	1	2	3	4
A	81.76	78.01	77.08	77.33
B	78.84	77.41	77.88	80.04
C	77.81	76.19	80.55	79.64

Table 4 Calculation results of average carbon content of concentrate

Factor	1	2	3	4
A	92.21	92.11	92.39	92.24
B	92.37	92.39	91.99	92.21
C	92.09	92.41	92.30	92.15

Table 5 Calculation results of average recovery of concentrate

Factor	1	2	3 ₃	4
A	84.44	80.39	79.59	79.80
B	81.37	80.03	80.28	82.56
C	80.34	78.75	83.17	81.98

Analyzing the data in table 3, 4 and 5, considering comprehensively, the optimal combination is A1B4C3, i.e. 40g. L⁻¹ of the pulp concentration, 300 g. t⁻¹ of the kerosene consumption and 125 g. t⁻¹ of the secondary octyl alcohol consumption are the optimal combination.

18. Page 3, line 22, “The mixture was stirred...”, Did the stirring process performed by a magnetic or a mechanical stirrer? Did the stirring process continue during the ultrasonic application? What was the stirring rate?

Answer:

This stirring is done by hand with a glass rod. The stirring process is neither performed by a magnetic stirrer nor by a mechanical stirrer. The stirring process did not continue during the ultrasonic application.

19. Page 3, line 23, “cooled to room temperature.” As you mentioned, ultrasound increased the pulp temperature which affects the flotation process in many ways. For instance, it can increase the activity of amine molecules (Gungoren, et al, 2017) Please explain your reasons for cooling the flotation pulp. Gungoren C., Ozdemir O., Ozkan S.G., 2017 Effects of temperature during ultrasonic conditioning in quartz-amine flotation. *Physicochem. Probl. Miner. Process.* 53(2), 2017, 687–698.

Answer:

The mixture was stirred and then ultrasonically treated. The ultrasonic treatment increased the pulp temperature which affects the flotation process in many ways. For instance, it can

increase the activity of amine molecules [30]. In order to eliminate the influence of temperature on graphite flotation, the ultrasonic treatment pulp was cooled to room temperature.

20. Page 3, line 25, “for 1 min” changed to “for a further min”

21. Page 3, line 26, “tailings of concentrates” is it “the concentrate loaded froth and tailings”?

Answer:

It is not “the concentrate loaded froth and tailings”. It is “collecting concentrate and tailings separately”. So, the sentence in the original text was changed to “Next, concentrates and tailings were collected, filtered, and dried separately. The dried samples were weighed to calculate the yield of the products.”

22. Page 3, line 37, $\varepsilon = (\beta \times \gamma_j / \alpha) \times 100\%$ (2) 100% This equals to 1. Please check this equation.

Answer:

This equation is correct, the aim of multiplied by 100% is to make the result percentage.

23. Page 3, line 43, “under” Maybe “at” is more convenient for ultrasonic powers. Please check throughout the manuscript.

Answer:

On the recommendation of the reviewer, the “under” was changed to “at” throughout the manuscript.

24. Page 3, line 47, . Only conventional flotation was performed in the absence of ultrasonic treatment. What did you mean with this sentence?

Answer:

The original paragraph was revised as follows: According to section 2.2, flotation tests were conducted by conventional flotation and ultrasonic-assisted flotation process at different ultrasound power (800, 1200, and 1600 W) and different ultrasonic treatment time separately. The yield, carbon content and recovery of flotation concentrate products were obtained during conventional and ultrasonic-assisted flotation process at different power and different ultrasonic treatment time. The results are presented in figure 2.

25. Page 3, line 50, “...was 2% higher than that...” It is better to mention the results before and after the ultrasonic treatment instead of giving the increment value.

Answer:

The original sentence was revised as follows: The yield, carbon content and recovery of the concentrate products of conventional flotation are 85.63%, 92.31% and 87.28%, respectively. It can be seen from figure 2 that the yield, carbon content and recovery of concentrate products obtained during ultrasonic-assisted flotation are higher in comparison to the concentrate products

of conventional flotation.

26. Page 3, line 53, “growth” changed to “increasing”

27. Page 3, line 54, “yield and recovery”. Please describe the difference between the “yield” and “recovery”

Answer:

“Yield” refers to the ratio of the separated product to the original ore. “Recovery rate” refers to the ratio of the separated pure mineral to the pure mineral in the original ore. The calculation of recovery rate should consider the purity of the product and the original ore.

28. Page 3, line 55, “There was the optimal flotation effect when the ultrasonic power was 1600 W.” Please consider revising this sentence.

Answer:

The original sentence “There was the optimal flotation effect when the ultrasonic power was 1600 W.” changed to “The most optimal flotation effect could be obtained when the ultrasonic power was 1600W.”

29. Page 3, line 56-57, Through the above-mentioned analysis, the carbon content of concentrates reached 95% after only one flotation step after ultrasonic treatment. This test verified that the introduction of ultrasonic waves can simplify the cleaning process of graphite.

You used a concentrate of a concentrator plant with 90.56% C content which was gone through several steps of “cleaning” phases. Can you make this comment?

Answer:

The concentrate with 90.56% C content which was gone through several steps of “cleaning” phases, explanations are given in section 2.1 and 3.1. Samples of 90.56% C content undergo two stages, first ultrasonic treatment, then flotation, and the carbon content of the flotation concentrate products reaches more than 95%.

30. Page 5, line 50-52, “Figures 2(a) and 2(b) separately present SEM images of untreated graphite and that treated by using ultrasonic waves for 5 min. Figure 2(a) shows that fine flaky bright components were found on the graphite surface, which were associated minerals and fine silt adsorbed onto the graphite surface.” Please consider revising this sentence.

Answer:

Since two graphs are added in this paper, the figure 2 changed to figure 4. The original sentence was modified to the following:

Figure 4a is the SEM images of raw graphite, and figure 4b is the SEM images of graphite after ultrasonic treatment for 5 min. Figure 4a shows that there are small pieces of bright components in the flaky graphite surface, which are associated minerals and fine mud adhering

to the graphite surface.

31. Page 5, line 54-57, “On the one hand, it revealed that the carbon content in graphite had been increased; on the other hand, it indicated that the hydrophobicity of graphite increased after ultrasonication.” Please consider revising this sentence.

Answer:

The original sentence was modified to “It can be supposed that ultrasonic treatment may increase the carbon content of graphite and enhance the hydrophobicity of graphite.”

32. Page 5, line 8-9, “Some 40 g of graphite was uniformly mixed with 500 mL of water, which was subjected to ultrasonic treatment for 5 min and then filtered and dried.” The explanation of the experimental methods needs to be given in Sect.2

Page 5, line 60, “During....”. The explanation of the experimental methods needs to be given in Sect.2. Please revise the sentences about the results of the effect of ultrasound on particle size of graphite.

Answer:

The following sections were moved to experimental section and some experimental procedures were rewritten according to the reviewer’s suggestion.

Page 5, line 8-12, section 3.2.1, the first paragraph, “40 g of graphite was uniformly mixed with 500 mL of water, which was ultrasonically treated for 5 min and then filtered and dried. Afterwards, the graphite with the same mass was taken and uniformly mixed with 500 mL of water and the mixture was filtered and dried. By using an MX2600 scanning electron microscope (SEM),the surface morphologies of graphite, in ultrasonicated, and untreated states were separately detected. ” was given in Sect.2

Page 5, line 60, section 3.2.2, the first paragraph, “40 g of graphite was uniformly mixed with 500 mL of water and then the mixture was ultrasonically treated for 5 min. Afterwards, wet screening was carried out. Graphite with the same mass, but untreated, was also wet screened.” was given in Sect.2

33. Page 6, line 4, ”particle-size grades “ the word “grade” is not convenient for particle size. Consider using “fraction” instead of “grade”.

Answer:

The word “fraction” was used instead of “grade” throughout the manuscript.

34. Page 6, line 6, ”graphite with larger particles“ changed to “larger sized graphite particles”.

35. Page 6, line 38, “die” Please consider to another word.

Answer:

The word “die” changed to “mould”

36. Page 6, line39, “round cakes” changed to “to form pellets”

37. Page 6, line40, ”from” changed to “on the surface the pellets”

38. Page 6, line43, ”circular graphite flake” changed to “pellet”

39. Page 6, line 44, “Table 2” Do the values in Table 2 representing the average values of three measurements?

Answer:

Yes, the values in table 2 represent the average values of three measurements.

40. Page 6, line 45, “Ultrasound treatment” I think the ultrasound was applied before pelleting. Please mention this in the paper.

Answer:

The sentence “Six graphite powder samples were prepared, one of which was not treated by ultrasound and the other five samples were treated by ultrasound for 2,3,4,5 and 6 min respectively.” was added to the first paragraph of section 3.3.1.

The sentence “In this study, the contact angle of the ultrasonic treatment graphite was compared with that of the raw graphite. Table 2 shows when the ultrasonic treatment time is 0 min, the contact angle of the raw graphite powder without ultrasonic treatment is 86.37° .” was added to the second paragraph of section 3.3.1.

41. Page 6, line54,”...strengthened the hydrophobicity...” The increase in the contact angle is just 2-3 degrees, which is under the experimental error margin. In addition, you can hardly make this comment with an increase of 2-3 degrees in the contact angle.

Answer:

We have carried out repeated experiments and obtained the same experimental law of the increase in the contact angle 2-3 degrees. In order to verify this rule, we have done the wetting heat experiment of graphite. The experimental results show that the hydrophobicity of graphite treated by ultrasound was improved. This is consistent with the experimental law of contact angle. It also proves that the law that the contact angle between graphite and water after ultrasonic treatment is larger than that without ultrasonic treatment is credible.

42. Page 7, line10-11, “...without undergoing ultrasonic treatment and in ultrasonically treated graphite, wet screening was separately carried out on the two types of graphite.” Please revise

Answer:

The original sentence was modified to “The ultrasonic treatment can change the particle size of graphite. In order to eliminate the influence of particle size change on the results of wetting heat, screening tests were carried out. To reveal the effects of ultrasonic treatment, the screening tests were conducted on conventionally and ultrasonically pre-treated graphite samples respectively.”

43. Page 7, line26-27, “As shown in Table 3, compared with graphite without undergoing ultrasonic treatment, the ultrasonically treated graphite at each grade always exhibited an increased wetting heat with kerosene” Please revise

Answer:

The original sentence was modified to “compared with original graphite, the ultrasonically treated graphite at each fraction exhibited an increased wetting heat with kerosene.”

44. Page 7, line39-40, ”Under X-ray irradiation, each matter emits a specific X-ray signal. A fluorescence spectrometer can capture these characteristic emissions to analyse wavelengths and determine the corresponding elements present” No need for this information.

Answer:

The sentence was deleted on the recommendation of the reviewer.

45. Page 7, line 41, “spectrometer” changed to “XRFspectrometer”.

46. Page 7, line45, ”measured” changed to ” measurement”.

47. Page 7, line46, “Concentrate by conventional flotation” changed to “Concentrate of conventional flotation”. “Concentrate by ultrasonic flotation” changed to “Concentrate of ultrasonic flotation”

48. Page 8, line5-6, “except for Ca whose elemental content decreased,” Please exclude this part.

Answer:

The sentence “.....however, except for Ca whose elemental content decreased, the amounts of all other metal elements assayed increased.” changed to “... however, the amounts of other metal elements except Ca increased.”

49. Page 8, line 7, “exerted an insignificant influence” changed to “was unsuccessful”

50. Page 8, line 16-17, “concentrates attained using flotation based on ultrasonic treatment decreased.” Consider revising.

Answer:

The sentence “... concentrates attained using flotation based on ultrasonic treatment decreased.” changed to “...concentrate products obtained during ultrasonic-assisted flotation

process decreased.”

51. Page 8, line 18,” flotation based on ultrasonic treatment” changed to “ultrasonic treatment flotation”.

52. Page 8, line39, “test” changed to “tests

53. Page 8, line 58, “References” Please check the Ref. formats.

Answer:

References were revised in accordance with the format required by the journal.

54. Some spaces between words are lost when word files are converted into PDF files are corrected.

For example:

Page 2, line 51, “Letmaheet al.” changed to “Letmahe et al.”

Page 3, line 9 “ina” changed to “in a”

Page 3, line 10 “massthere” changed to “mass there”

Page 3, line25, “octanolinto” changed to “octanol into”

Page 3, line 22, “wasstirred” changed to “was stirred”

Appendix C**ROYAL SOCIETY
OPEN SCIENCE****Enhancement of flaky graphite cleaning by ultrasonic
treatment**

Journal:	Royal Society Open Science
Manuscript ID	RSOS-191160.R1
Article Type:	Research
Date Submitted by the Author:	18-Sep-2019
Complete List of Authors:	Kang, Wenze; Heilongjiang University of Science and Technology, Institute of mining engineering Li, Huijian; Heilongjiang University of Science and Technology, Institute of mining engineering
Subject:	Materials science < ENGINEERING AND TECHNOLOGY
Keywords:	Ultrasound, Flotation, Surface Properties, Graphite
Subject Category:	Engineering

Author-supplied statements

Relevant information will appear here if provided.

Ethics

Does your article include research that required ethical approval or permits?:

This article does not present research with ethical considerations

Statement (if applicable):

CUST_IF_YES_ETHICS :No data available.

Data

It is a condition of publication that data, code and materials supporting your paper are made publicly available. Does your paper present new data?:

My paper has no data

Statement (if applicable):

CUST_IF_YES_DATA :No data available.

Conflict of interest

I/We declare we have no competing interests

Statement (if applicable):

CUST_STATE_CONFLICT :No data available.

Authors' contributions

This paper has multiple authors and our individual contributions were as below

Statement (if applicable):

Wenze Kang carried out the flotation lab work, participated in data analysis, participated in the design of the study, conceived of the study, designed the study, coordinated the study and drafted the manuscript. Huijian Li carried out the statistical analyses and collected field data. All authors gave final approval for publication.

Reply for the comments of reviewer 1**General comments of reviewer 1:**

Slight modifications in test and figure.9 recommended as indicated in paper.

Minor revision is recommended. This manuscript may be accepted after fulfilling the following comments. Written English of the manuscript needs to be improved for better clarity.

According to the comments of review 1, we answered the questions and made modifications as following:

1. Authors may add some lines or texts to the revised manuscript precisely indicating the importance of this research, its novelty and industrial relevance.

Answer:

The emphasis of this paper is on the enhancement of graphite flotation by ultrasound. The aim is to simplify the graphite cleaning process and promote the efficient development of graphite industry. In this paper, the flow-sheet of conventional flotation and the flow- sheet of ultrasonic treatment flotation are added to show that intensifying graphite flotation by ultrasonic treatment can shorten the graphite cleaning process. Technological process flow-sheet of the graphite concentrator in Heilongjiang, China is shown in figure 1. Technological graphite flotation processing flow-sheet using ultrasonic treatment is shown in the figure 3.

Figure.1 Technological processing flow-sheet of the Luobei graphite concentrator

Figure.3 Technological graphite flotation processing flow-sheet using ultrasonic treatment

2. Include these papers in the introduction section to support your work strongly:

DOI: <https://doi.org/10.1016/j.ultsonch.2019.04.033>—

DOI: <https://doi.org/10.1016/j.ultsonch.2018.08.016>—

DOI: <https://doi.org/10.1021/acs.energyfuels.9b01543>—

Answer:

These three papers have been added to the introduction as references 21, 22 and 29.

3. Include few more literatures particularly on the work published on flake graphite and discuss how the published work is different than your work. You can use this paper to have some preliminary idea: DOI: <https://doi.org/10.1016/j.ultsonch.2019.04.033>

Answer:

The paper recommended by the reviewer was cited in the introduction and listed in the

references. The following sentences were added to the introduction. Santosh DebBarma et al.[29] investigated the feasibility of low-frequency ultrasound in enhancing the flotability of flaky graphite from low-grade graphite ore. Their results showed that the yield, fixed carbon content and percentage recovery of the flotation concentrate products increased significantly under ultrasonic-assisted flotation.

The emphasis of my study is on the enhancement of graphite flotation by ultrasound. The aim is to simplify the graphite cleaning process and promote the efficient development of graphite industry.

4. Comparison between conventional and ultrasonicated flotation may be provided using recovery, yield and fixed carbon in the abstract.

Answer:

In the abstract section, the following paragraph was added according to the reviewer's suggestion.

“The yield, carbon content and recovery of flotation concentrate were 91.46%, 95.17% and 96.12% after ultrasonic treatment for 4 minutes with ultrasound power 1600W, which were 5.83%, 2.86% and 8.84% higher than conventional flotation, respectively.”

5. In section 2.2, authors may provide reference against the “orthogonal experiments” for better clarity of the optimization process being used.

Answer:

We did the orthogonal experiment. The process of the experiment was as follows. It was limited to the length of the article and was not put in the text.

On the basis of single factor test, the main factors such as pulp concentration (A), dosage of kerosene (B) and dosage of secondary octyl alcohol (C) were selected. Design orthogonal experiments to determine the optimum parameters. The orthogonal experimental design is shown in table 1 and the flotation experimental results are shown in table 2.

On the basis of single factor test, the main factors such as pulp concentration (A), dosage of kerosene (B) and dosage of secondary octyl alcohol (C) were selected. Design orthogonal experiments to determine the optimum parameters. The orthogonal experimental design is shown in table 1 and the flotation experimental results are shown in table 2.

Table 1 Orthogonal experiment design

	A	B	C
level	/g·L ⁻¹	/g·t ⁻¹	/g·t ⁻¹
1	40	150	75
2	50	200	100
3	60	250	125

4 70 300 150

Table 2 Flotation experimental results

number	A	B	C	yield	carbon	recovery
	/g·L ⁻¹	/g·t ⁻¹	/g·t ⁻¹	/%	content /%	
A1B1C1	40	150	75	77.24	91.75	79.45
A1B2C2	40	200	100	78.31	92.76	81.22
A1B3C3	40	250	125	85.09	92.27	88.06
A1B4C4	40	300	150	86.41	92.08	89.05
A2B1C2	50	150	100	77.79	92.55	80.51
A2B2C1	50	200	75	78.36	92.22	81.01
A2B3C4	50	250	150	77.24	91.58	79.00
A2B4C3	50	300	125	78.67	92.10	81.05
A3B1C3	60	150	125	81.23	92.66	83.93
A3B2C4	60	200	150	75.80	92.39	78.28
A3B3C1	60	250	75	75.93	92.13	78.47
A3B4C2	60	300	100	75.38	92.37	77.69
A4B1C4	70	150	150	79.12	92.53	81.57
A4B2C3	70	200	125	77.19	92.19	79.62
A4B3C2	70	250	100	73.27	91.97	75.59
A4B4C1	70	300	75	79.72	92.27	82.43

According to the flotation experiment results in table 2, the average values of concentrate yield, carbon content and recovery of products with different factors and levels are calculated. The results are shown in table 3, Table 4 and Table 5.

Table 3 Calculation results of average yield of concentrate

Factor	1	2	3	4
A	81.76	78.01	77.08	77.33
B	78.84	77.41	77.88	80.04
C	77.81	76.19	80.55	79.64

Table 4 Calculation results of average carbon content of concentrate

Factor	1	2	3	4
A	92.21	92.11	92.39	92.24

B	92.37	92.39	91.99	92.21
C	92.09	92.41	92.30	92.15

Table5 Calculation results of average recovery of concentrate

Factor	1	2	3 ₃	4
A	84.44	80.39	79.59	79.80
B	81.37	80.03	80.28	82.56
C	80.34	78.75	83.17	81.98

Analyzing the data in table 3, 4 and 5, considering comprehensively, the optimal combination is A1B4C3, i.e. 40g. L⁻¹ of the pulp concentration, 300 g. t⁻¹ of the kerosene consumption and 125 g. t⁻¹ of the secondary octyl alcohol consumption are the optimal combination.

6. Authors have mentioned that, at higher power (1600 W), yield and recovery were found to be higher than those of 800 and 1200 W. Now the major concern here is its economic aspects since the higher power will consume more energy. Kindly justify this.

Answer:

For example, when the ultrasound treatment time is 4 min, the concentrate yield of 800W, 1200W and 1600W is 86.56%, 89.12% and 91.46% respectively.

200 g of original sample can be processed for 4 min in JAC-5500 Ultrasound Generator every time. It's 0.5 yuan RMB per kilowatt hour. The electricity charges for 800W, 1200W and 1600W of a ton of original sample are 133, 200 and 267 yuan RMB respectively. The electricity charges for 1600W ultrasound are 134 and 67 yuan RMB more than that of 800W and 1200W.

The market price of graphite products with 95% carbon content is 4500 yuan RMB per ton. Because of the high concentrate yield obtained by 1600W ultrasound, the income of treating 1 ton of original 1600W ultrasound is 245 yuan RMB and 117 yuan RMB more than that of 800W and 1200W respectively.

The income of a ton of original sample with 1600W ultrasonic treatment is 111 yuan RMB and 50 yuan RMB more than that of 800W and 1200W respectively. Consequently, the economic benefits of 1600W ultrasound are better.

7. What are the values of yield, recovery and fixed carbon content of conventional flotation? Please compare them for more clarity in section 3.1.

Answer:

In section 3.1, the second paragraph, the following paragraph was added.

Points of the ultrasonic treatment time 0 min in the figures 2(a), (b), and (c) represent the products parameters obtained by conventional flotation. The yield, carbon content and recovery of the concentrate products of conventional flotation are 85.63%, 92.31% and 87.28%, respectively.

8. In section 3.4, the elemental compositions of each feed and products varies differently. For
instance, “Al and Fe” in the concentrate of conventional flotation is higher than its original sample.
Similar trends can also be seen in concentrate of ultrasonicated flotation. Kindly comment on this.

**Answer:**

Fluorescence spectrometry is used to determine the relative content of elements in samples.
The content of elements in each sample is calculated by 100%. After ultrasonic treatment, some
minerals may be separated from the sample, resulting in the reduction of some elements in the
sample. As the content of some elements decreases, the content of other elements is increased
relatively. For example, the content of Si, S and Ca after conventional flotation is lower, while the
content of Al, K, Fe and Ag is higher than that in original samples.

9. Contact angle on ultrasonicated graphite concentrate may be compared with those without
ultrasonicated one (raw sample).

**Answer:**

In this study, the contact angle of the ultrasonic treatment graphite concentrate was compared
with those without ultrasonic treatment one (raw graphite). Table 2 shows the contact angle of the
ultrasonic treatment graphite concentrate in different ultrasonic treatment time and those without
ultrasonicated one. When the ultrasonic treatment time is 0 min, i.e., without ultrasonic treatment,
the contact angle of graphite powder is 86.37°.

The sentence “Six graphite powder samples were prepared, one of which was not treated by
ultrasound and the other five samples were treated by ultrasound for 2,3,4,5 and 6 min
respectively.” was added to the first paragraph of section 3.3.1.

The sentence “In this study, the contact angle of the ultrasonic treatment graphite was compared
with that of the raw graphite. Table 2 shows when the ultrasonic treatment time is 0 min, i.e.,
without ultrasonic treatment, the contact angle of the raw graphite pellet is 86.37°.” was added to
the second paragraph of section 3.3.1.

10. I am confused if the reference no. 32 can be used here as a reference. This seems to be the
same unpublished manuscript.

**Answer:**

According to open data policies of Royal Society Open Science, it is a condition of
submission and publication that all supporting data for published articles are made freely available
either as supplementary information or preferably via an appropriate repository.

This link will only be active while my manuscript is under review. It will allow editors and
reviewers to access my data while my manuscript is under review.

I may include the temporary review link in my manuscript now for reviewers, but that it will no
longer work when my data are publicly archived. If/when my manuscript is accepted for

publication, and my data are publicly archived, the link to access my data will change to the
permanent DOI.

11. Written English of the manuscript needs to be improved for better clarity.

**Answer:**

We noticed some spaces between words are lost when word files are converted into PDF files
and corrected them.

Written English of the manuscript has been revised for better clarity especially the abstract
section. The abstract revised as follows:

**Abstract:** In this study, the aim is to simplify the graphite cleaning processing. In order to achieve
flotation for graphite effectively, ultrasonic treatment was utilized as a pre-treatment technique.
Flotation tests were conducted using for different ultrasound power and ultrasonic treatment time.
The influences of ultrasonic treatment on particle sizes, morphologies, wettability, content of
surface elements and on the flotation effect of flaky graphite were investigated. The results of
ultrasonic treatment for graphite flotation were compared to the results of conventional flotation.
The results showed that ultrasonic treatment not only changed the size of flaky graphite but also
eliminated impurities on the graphite surface. Additionally, ultrasonic treatment improved the
hydrophobicity of graphite. The effects of ultrasound were observed that it can remove not only
silicate impurities but also most other metal impurities. The yield, carbon content and recovery of
flotation concentrate were 91.46%, 95.17% and 96.12% after ultrasonic treatment for 4 minutes
with ultrasound power 1600W, which were 5.83%, 2.86% and 8.84% higher than that of
conventional flotation, respectively. The graphite after ultrasonic treatment was conducted only
one times flotation, the carbon content in concentrates products had reached 95%.
This study indicates that intensifying graphite flotation by ultrasonic treatment can shorten the
graphite cleaning process.

Reply for the comments of reviewer 2**General comments of reviewer 2:**

There are several unclear points and many language errors in the paper which should be answered and corrected before publication. Therefore, in my opinion, the paper needs major revisions.

According to the comments of review 2, we answered the questions and made modifications as following:

1. Page 2, line 4, ultrasonication There are many different terms for ultrasonic treatment in the paper(with/without/based on ultrasonic treatment or ultrasonic waves, ultrasonication, application of ultrasonic treatment, unultrasonic treatment) This has a significant negative effect on the understandability of the paper. Please chose one of them for "ultrasonic treatment" and one other for the "no ultrasonic treatment" and use them throughout the paper.

Answer:

According to the reviewer's suggestion, we have revised the English for ultrasound and non-ultrasound treatment throughout the paper.

2. Page 2, line 18, 4 min the power of ultrasound can be added in the abstract.

Answer:

In section abstract, the following paragraph was added according to the reviewer's suggestion.

The yield, carbon content and recovery of flotation concentrate were 91.46%, 95.17% and 96.12% after ultrasonic treatment for 4 minutes with ultrasound power 1600W, which were 5.83%, 2.86% and 8.84% higher than conventional flotation, respectively.

3. Page 2, line 1, Keywords: "Graphite" should be added to the keywords.

Answer:

The words "Graphite" was added to the keywords according to the reviewer comment.

4. Page 2, line 29, Requirements: citation required

Answer:

The following references were added in the first paragraph in the introduction section.

Reference 1. Chelgani SC, Rudolph M, Kratzsch R, Sandmann D, Gutzmer J. 2016 A review of graphite beneficiation techniques, Miner. Process. Extr. Metall. Rev. 37, 58–68.

Doi:10.1080/08827508.2015.1115992

Reference 2. Vasumathi N, Vijaya Kumar TV, Ratchambigai S, Subba Rao S, Bhaskar Raju G, 2015, Flotation studies on low grade graphite ore from eastern India. Int. J. Min. Sci. Technol. 25, 415–420 DOI: 10.3969/j.issn.2095-2686.2015.03.013

Reference 3.W. Peng, Y. Qiu, L. Zhang, J. Guan, S. Song, 2017Increasing the fine flaky
graphite recovery in flotation via a combined multiple treatments technique of middlings,
Minerals 7(11): 208.DOI:
10.3390/min7110208

5. Page 2, line 29, “in the range of5~15%”, is it 5~15%C?

**Answer:**

Yes, it is. The sentence changed to “the range of carbon content is 5~15%”

6. Page 2, line 40 “application of ultrasonic treatment”, “application” and “treatment” are the
words with almost the same meaning

**Answer:**

The word “application” was deleted.

7. Page 2, line 42, “The research”: which one?

**Answer:**

“The research” changed to “Existing studies”. It is not a single research, but it is several
researches such as references 4-11.

8. Page 2, line 44, “Scholars” changed to “Some researchers investigated”.

9. Page 2, line 47, “Scholars” changed to “Some other researchers studied”.

10. Page 2, line 49, “The research”: which one?

**Answer:**

“The research” changed to “Their researches”. It is not a single research, but It is several
researches such as references 23-27.

11. Page 2, line53, “so as to increase the effect thereof,” incomprehensible sentence

**Answer:**

Original sentence “After conducting ultrasonic treatment, reagents were more uniformly
distributed in suspension so as to increase the effect thereof, thus increasing the purity of
concentrates and decreasing the ash content of concentrates.” was deleted and changed to
“Ultrasonic treatment can improve the effectiveness of a reagent due to a more uniform
distribution in the suspension.”

12. Page 2, line 56, “sparse”: There are many studies on the ultrasonic application on graphite.
Please specify the difference of your study from these studies.

**Answer:**

Previous researchers have made remarkable achievements in enhancing minerals flotation by
ultrasound, but the effect of ultrasonic treatment on graphite cleaning process is studied less.
To develop a method for shortening the graphite cleaning process, the influences of ultrasonic
treatment on changes of particle sizes, morphologies, wettability, content of surface elements of
flaky graphite, and on the flotation effect of flaky graphite were explored in this paper. The
emphasis of this paper is on the enhancement of graphite flotation by ultrasound. The aim is to
simplify the graphite cleaning process and promote the efficient development of graphite industry.

13. Page 2, line 58, “explored...” in this study?

**Answer:**

Yes, it is. So, the sentence changed to “...this cleaning process was explored in this study.”

14. Page 3, line 7 “taken” changed to “were taken”

15. Page 3, line 8, “cleaning”, “what is the method for cleaning the graphite in this concentrator?”

**Answer:**

For illustration the method for cleaning the graphite in this concentrator, the section 2.1
changed as following:

The test samples were taken from a graphite concentrator in Luobei, Heilongjiang Province,
China. The beneficiation process of the concentrator consisted of crushing, coarse and rougher,
nine stages regrinding and ten stages cleaning as shown in figure1. The flotation method is the
method for cleaning the graphite in this concentrator.

16. Page 3, line 14, “ applied” changed to “employed”

17. Page 3, line 17, “determined” Why did not you give the results of these optimization
experiments in the paper?

**Answer:**

We did the orthogonal experiment. The process of the experiment was as follows. It was
limited to the length of the article and was not put in the text.

On the basis of single factor test, the main factors such as pulp concentration (A), dosage of
kerosene (B) and dosage of secondary octyl alcohol (C) were selected. Design orthogonal
experiments to determine the optimum parameters. The orthogonal experimental design is shown
in Table 1 and the flotation experimental results are shown in Table 2.

On the basis of single factor test, the main factors such as pulp concentration (A), dosage of
kerosene (B) and dosage of secondary octyl alcohol (C) were selected. Design orthogonal
experiments to determine the optimum parameters. The orthogonal experimental design is shown
in Table 1 and the flotation experimental results are shown in Table 2.

Table 1 Orthogonal experiment design

	A	B	C
level	/g·L-1	/g·t-1	/g·t-1
1	40	150	75
2	50	200	100
3	60	250	125
4	70	300	150

Table 2 Flotation experimental results

number	A	B	C	yield	carbon	recovery
	/g·L-1	/g·t-1	/g·t-1	/%	content /%	
A1B1C1	40	150	75	77.24	91.75	79.45
A1B2C2	40	200	100	78.31	92.76	81.22
A1B3C3	40	250	125	85.09	92.27	88.06
A1B4C4	40	300	150	86.41	92.08	89.05
A2B1C2	50	150	100	77.79	92.55	80.51
A2B2C1	50	200	75	78.36	92.22	81.01
A2B3C4	50	250	150	77.24	91.58	79.00
A2B4C3	50	300	125	78.67	92.10	81.05
A3B1C3	60	150	125	81.23	92.66	83.93
A3B2C4	60	200	150	75.80	92.39	78.28
A3B3C1	60	250	75	75.93	92.13	78.47
A3B4C2	60	300	100	75.38	92.37	77.69
A4B1C4	70	150	150	79.12	92.53	81.57
A4B2C3	70	200	125	77.19	92.19	79.62
A4B3C2	70	250	100	73.27	91.97	75.59
A4B4C1	70	300	75	79.72	92.27	82.43

According to the flotation experiment results in Table 2, the average values of concentrate yield, carbon content and recovery of products with different factors and levels are calculated. The results are shown in Table 3, Table 4 and Table 5.

Table 3 Calculation results of average yield of concentrate

Factor	1	2	3	4
A	81.76	78.01	77.08	77.33
B	78.84	77.41	77.88	80.04
C	77.81	76.19	80.55	79.64

Table 4 Calculation results of average carbon content of concentrate

Factor	1	2	3	4
A	92.21	92.11	92.39	92.24
B	92.37	92.39	91.99	92.21
C	92.09	92.41	92.30	92.15

Table 5 Calculation results of average recovery of concentrate

Factor	1	2	3 ₃	4
A	84.44	80.39	79.59	79.80
B	81.37	80.03	80.28	82.56
C	80.34	78.75	83.17	81.98

Analyzing the data in table 3, 4 and 5, considering comprehensively, the optimal combination is A1B4C3, i.e. 40g. L⁻¹ of the pulp concentration, 300 g. t⁻¹ of the kerosene consumption and 125 g. t⁻¹ of the secondary octyl alcohol consumption are the optimal combination.

18. Page 3, line 22, “The mixture was stirred...”, Did the stirring process performed by a magnetic or a mechanical stirrer? Did the stirring process continue during the ultrasonic application? What was the stirring rate?

Answer:

This stirring is done by hand with a glass rod. The stirring process is neither performed by a magnetic stirrer nor by a mechanical stirrer. The stirring process did not continue during the ultrasonic application.

19. Page 3, line 23, “cooled to room temperature.” As you mentioned, ultrasound increased the pulp temperature which affects the flotation process in many ways. For instance, it can increase the activity of amine molecules (Gungoren, et al, 2017) Please explain your reasons for cooling the flotation pulp. Gungoren C., Ozdemir O., Ozkan S.G., 2017 Effects of temperature during ultrasonic conditioning in quartz-amine flotation. *Physicochem. Probl. Miner. Process.* 53(2), 2017, 687–698.

Answer:

The mixture was stirred and then ultrasonically treated. The ultrasonic treatment increased the pulp temperature which affects the flotation process in many ways. For instance, it can

increase the activity of amine molecules [30]. In order to eliminate the influence of temperature on
graphite flotation, the ultrasonic treatment pulp was cooled to room temperature.

20. Page 3, line 25, “for 1 min” changed to “for a further min”

21. Page 3, line 26, “tailings of concentrates” is it “the concentrate loaded froth and tailings”?

**Answer:**

It is not “the concentrate loaded froth and tailings”. It is “collecting concentrate and tailings
separately”. So, the sentence in the original text was changed to “Next, concentrates and tailings
were collected, filtered, and dried separately. The dried samples were weighed to calculate the
yield of the products.”

22. Page 3, line 37, $\varepsilon = (\beta \times \gamma_j / \alpha) \times 100\%$ (2) 100% This equals to 1. Please check this equation.

**Answer:**

This equation is correct, the aim of multiplied by 100% is to make the result percentage.

23. Page 3, line 43, “under” Maybe “at” is more convenient for ultrasonic powers. Please check
throughout the manuscript.

**Answer:**

On the recommendation of the reviewer, the “under” was changed to “at” throughout the
manuscript.

24. Page 3, line 47, . Only conventional flotation was performed in the absence of ultrasonic
treatment. What did you mean with this sentence?

**Answer:**

The original paragraph was revised as follows: According to section 2.2, flotation tests were
conducted by conventional flotation and ultrasonic-assisted flotation process at different
ultrasound power (800, 1200, and 1600 W) and different ultrasonic treatment time separately.
The yield, carbon content and recovery of flotation concentrate products were obtained during
conventional and ultrasonic-assisted flotation process at different power and different ultrasonic
treatment time. The results are presented in figure 2.

25. Page 3, line 50, “...was 2% higher than that...” It is better to mention the results before and
after the ultrasonic treatment instead of giving the increment value.

**Answer:**

The original sentence was revised as follows: The yield, carbon content and recovery of the
concentrate products of conventional flotation are 85.63%, 92.31% and 87.28%, respectively. It
can be seen from figure 2 that the yield, carbon content and recovery of concentrate products
obtained during ultrasonic-assisted flotation are higher in comparison to the concentrate products

of conventional flotation.

26. Page 3, line 53, “growth” changed to “increasing”

27. Page 3, line 54, “yield and recovery”. Please describe the difference between the “yield” and
“recovery”

**Answer:**

“Yield” refers to the ratio of the separated product to the original ore. “Recovery rate” refers
to the ratio of the separated pure mineral to the pure mineral in the original ore. The calculation of
recovery rate should consider the purity of the product and the original ore.

28. Page 3, line 55, “There was the optimal flotation effect when the ultrasonic power was 1600
20 W.” Please consider revising this sentence.

**Answer:**

The original sentence “There was the optimal flotation effect when the ultrasonic power was
1600 W.” changed to “The most optimal flotation effect could be obtained when the ultrasonic
power was 1600W.”

29. Page 3, line 56-57, Through the above-mentioned analysis, the carbon content of concentrates
reached 95% after only one flotation step after ultrasonic treatment. This test verified that the
introduction of ultrasonic waves can simplify the cleaning process of graphite.

You used a concentrate of a concentrator plant with 90.56% C content which was gone through
several steps of “cleaning” phases. Can you make this comment?

**Answer:**

The concentrate with 90.56% C content which was gone through several steps of “cleaning”
phases, explanations are given in section 2.1 and 3.1. Samples of 90.56% C content undergo two
stages, first ultrasonic treatment, then flotation, and the carbon content of the flotation concentrate
products reaches more than 95%.

30. Page 5, line 50-52, “Figures 2(a) and 2(b) separately present SEM images of untreated
graphite and that treated by using ultrasonic waves for 5 min. Figure 2(a) shows that fine flaky
bright components were found on the graphite surface, which were associated minerals and fine
silt adsorbed onto the graphite surface.” Please consider revising this sentence.

**Answer:**

Since two graphs are added in this paper, the figure 2 changed to figure 4. The original
sentence was modified to the following:

Figure 4a is the SEM images of raw graphite, and figure 4b is the SEM images of graphite
after ultrasonic treatment for 5 min. Figure 4a shows that there are small pieces of bright
components in the flaky graphite surface, which are associated minerals and fine mud adhering

to the graphite surface.

31. Page 5, line 54-57, “On the one hand, it revealed that the carbon content in graphite had been
increased; on the other hand, it indicated that the hydrophobicity of graphite increased after
ultrasonication.” Please consider revising this sentence.

**Answer:**

The original sentence was modified to “It can be supposed that ultrasonic treatment may
increase the carbon content of graphite and enhance the hydrophobicity of graphite.”

32. Page 5, line 8-9, “Some 40 g of graphite was uniformly mixed with 500 mL of water, which
was subjected to ultrasonic treatment for 5 min and then filtered and dried.” The explanation of
the experimental methods needs to be given in Sect.2

Page 5, line 60, “During....”. The explanation of the experimental methods needs to be given in
Sect.2. Please revise the sentences about the results of the effect of ultrasound on particle size of
graphite.

**Answer:**

The following sections were moved to experimental section and some experimental
procedures were rewritten according to the reviewer’s suggestion.

Page 5, line 8-12, section 3.2.1, the first paragraph, “40 g of graphite was uniformly mixed
with 500 mL of water, which was ultrasonically treated for 5 min and then filtered and dried.
Afterwards, the graphite with the same mass was taken and uniformly mixed with 500 mL of
water and the mixture was filtered and dried. By using an MX2600 scanning electron microscope
(SEM),the surface morphologies of graphite, in ultrasonicated, and untreated states were
separately detected.” was given in Sect.2

Page 5, line 60, section 3.2.2, the first paragraph, “40 g of graphite was uniformly mixed with
500 mL of water and then the mixture was ultrasonically treated for 5 min. Afterwards, wet
screening was carried out. Graphite with the same mass, but untreated, was also wet screened.”
was given in Sect.2

33. Page 6, line 4, ”particle-size grades “ the word “grade” is not convenient for particle size.
Consider using “fraction” instead of “grade”.

**Answer:**

The word “fraction” was used instead of “grade” throughout the manuscript.

34. Page 6, line 6, ”graphite with larger particles“ changed to “larger sized graphite particles”.

35. Page 6, line 38, “die” Please consider to another word.

**Answer:**

The word “die” changed to “mould”

36. Page 6, line39, “round cakes” changed to “to form pellets”

37. Page 6, line40, ”from” changed to “on the surface the pellets”

38. Page 6, line43, ”circular graphite flake” changed to “pellet”

39. Page 6, line 44, “Table 2” Do the values in Table 2 representing the average values of three
measurements?

**Answer:**

Yes, the values in table 2 represent the average values of three measurements.

40. Page 6, line 45, “Ultrasound treatment” I think the ultrasound was applied before pelleting.
Please mention this in the paper.

**Answer:**

The sentence “Six graphite powder samples were prepared, one of which was not treated by
ultrasound and the other five samples were treated by ultrasound for 2,3,4,5 and 6 min
respectively.” was added to the first paragraph of section 3.3.1.

The sentence “In this study, the contact angle of the ultrasonic treatment graphite was
compared with that of the raw graphite. Table 2 shows when the ultrasonic treatment time is 0 min,
the contact angle of the raw graphite powder without ultrasonic treatment is 86.37°.” was added to
the second paragraph of section 3.3.1.

41. Page 6, line54,”...strengthened the hydrophobicity...” The increase in the contact angle is
just 2-3 degrees, which is under the experimental error margin. In addition, you can hardly make
this comment with an increase of 2-3 degrees in the contact angle.

**Answer:**

We have carried out repeated experiments and obtained the same experimental law of the
increase in the contact angle 2-3 degrees. In order to verify this rule, we have done the wetting
heat experiment of graphite. The experimental results show that the hydrophobicity of graphite
treated by ultrasound was improved. This is consistent with the experimental law of contact
angle. It also proves that the law that the contact angle between graphite and water after
ultrasonic treatment is larger than that without ultrasonic treatment is credible.

42. Page 7, line10-11, “...without undergoing ultrasonic treatment and in ultrasonically treated
graphite, wet screening was separately carried out on the two types of graphite.” Please revise

**Answer:**

The original sentence was modified to “The ultrasonic treatment can change the particle size
of graphite. In order to eliminate the influence of particle size change on the results of wetting heat,
screening tests were carried out. To reveal the effects of ultrasonic treatment, the screening tests
were conducted on conventionally and ultrasonically pre-treated graphite samples respectively.”

43. Page 7, line26-27, “As shown in Table 3, compared with graphite without undergoing
ultrasonic treatment, the ultrasonically treated graphite at each grade always exhibited an
increased wetting heat with kerosene” Please revise

**Answer:**

The original sentence was modified to “compared with original graphite, the ultrasonically
treated graphite at each fraction exhibited an increased wetting heat with kerosene.”

44. Page 7, line39-40, ”Under X-ray irradiation, each matter emits a specific X-ray signal. A
fluorescence spectrometer can capture these characteristic emissions to analyse wavelengths and
determine the corresponding elements present” No need for this information.

**Answer:**

The sentence was deleted on the recommendation of the reviewer.

45. Page 7, line 41, “spectrometer” changed to “XRFspectrometer”.

46. Page 7, line45, ”measured” changed to ” measurement”.

47. Page 7, line46, “Concentrate by conventional flotation” changed to “Concentrate of
conventional flotation”. “Concentrate by ultrasonic flotation” changed to “Concentrate of
ultrasonic flotation”

48. Page 8, line5-6, “except for Ca whose elemental content decreased,” Please exclude this part.

**Answer:**

The sentence “.....however, except for Ca whose elemental content decreased, the amounts
of all other metal elements assayed increased.” changed to “... however, the amounts of other
metal elements except Ca increased.”

49. Page 8, line 7, “exerted an insignificant influence” changed to “was unsuccessful”

50. Page 8, line 16-17, “concentrates attained using flotation based on ultrasonic treatment
decreased.” Consider revising.

**Answer:**

The sentence “... concentrates attained using flotation based on ultrasonic treatment
decreased.” changed to “...concentrate products obtained during ultrasonic-assisted flotation

process decreased.”

51. Page 8, line 18, “flotation based on ultrasonic treatment” changed to “ultrasonic treatment
flotation”.

52. Page 8, line39, “test” changed to “tests

53. Page 8, line 58, “References” Please check the Ref. formats.

**Answer:**

References were revised in accordance with the format required by the journal.

54. Some spaces between words are lost when word files are converted into PDF files are
corrected.

For example:

Page 2, line 51, “Letmaheet al.” changed to “Letmahe et al.”

Page 3, line 9 “ina” changed to “in a”

Page 3, line 10 “massthere” changed to “mass there”

Page 3, line25, “octanolinto” changed to “octanol into”

Page 3, line 22, “wasstirred” changed to “was stirred”

[revised manuscript text omitted]

22 (a) Surface morphology of graphite without ultrasonic treatment

40 (b) Surface morphology of graphite with ultrasonic treatment

41 Figure 4 Surface topography of graphite

Table 1 Test results of graphite screening (%)

Particle-size fraction/mm	Original graphite		Ultrasonic treatment graphite		Fraction change rate	
	Yield	Carbon content	Yield	Carbon content	Yield	Carbon content
+0.25	1.44	94.76	0.77	95.60	-46.53	0.89
0.25-0.15	9.71	94.15	3.59	95.27	-63.03	1.19
0.15-0.074	37.32	92.93	20.19	94.35	-45.90	1.54
0.074-0.045	8.34	91.55	8.59	92.38	3.00	0.91
-0.045	43.19	89.14	66.85	90.43	54.78	1.45

Table 2 Contact angle measured after graphite pressed into pellets

Ultrasonic treatment time/min	0	2	3	4	5	6
Contact angle /°	86.37	87.15	88.11	89.34	89.61	89.56

Table 3 The wetting heat of graphite and kerosene at different particle size ($\text{J} \cdot \text{g}^{-1}$)

Particle-size fractions/mm	Original graphite	Ultrasound-treated graphite
0.25-0.15	0.071	0.148
0.15-0.74	0.083	0.300
0.074-0.045	0.307	0.326
-0.045	0.489	0.561

Table 4 Measurement results of main elements in graphite%

Element	Original sample/%	Concentrate of conventional flotation/%	Ultrasonically treated sample/%	Concentrate of ultrasonic flotation/%
Al	6.398	6.617	8.543	5.5
Si	18.109	15.652	22.06	13.895
P	0.423	0.5	1.057	1.743
S	7.374	5.262	5.758	6.716
K	7.541	8.48	12.497	7.424
Ca	9.085	5.33	6.9	5.23
Fe	41.749	47.051	31.51	46.327
Cl	1.639	2.003	2.313	2.639
Ag	0.781	0.942	0.541	1.473

Appendix D

Reply for the comments of reviewer

According to the comments of reviewer, we answered the questions and made modifications as following:

1. Page 3, line 18, "Previous researchers have made remarkable achievements in enhancing minerals flotation by ultrasound, but the effect of ultrasonic treatment on graphite cleaning process is studied less." Requires citation.

Answer:

The following references were added in the section.

Previous researchers have made remarkable achievements in enhancing minerals flotation by ultrasound [21, 36], but the effect of ultrasonic treatment on graphite cleaning process is studied less [29].

2. Page 8, line 54, in some values there are spaces between the numbers and the units while there are not. Please use one of them.

Answer:

There are spaces between numbers and units in all values.

3. Some lost spaces between words are corrected and the font inconsistency in the text has been modified.

For example:

Page 2, line 59, "treatmentflotationfor" changed to "treatment flotation for".

Page 3, line 8 "et al.[28]" changed to "*et al.* [28]".

Page 3, line 11, "Santosh DebBarma et al.[29]" changed to "Barma *et al.* [29]"

Page 3, line 40, "30Kg" changed to "30 kg".

Page 3, line 53, "experiment" changed to "experiments".

Page 3, line 55, "1200and" changed to "1200, and".

Page 3, line 58, "test" changed to "experiments".

Page 4, line 51, "content and" changed to "content, and".

Page 4, line 54, "infigure" changed to "in figure".

Page 4, line 59, "92.31% and" changed to "92.31%, and".

Page 4, line 59, "figure2that" changed to "figure 2 that".

Page 5, line 5, "91.46%, 95.17% and 96.12%" changed to "91.46%, 95.17%, and 96.12%".

Page 5, line 6, "5.83%, 2.86% and 8.84%" changed to "5.83%, 2.86%, and 8.84%".

Page 5, line 7, "...respectively.On..." changed to "...respectively. On...".

Page 7, line 48 and 49, "Figure 4a" changed to "Figure 4 (a)".

Page 7, line 53, "mudadsorbed" changed to

Page 8, line 14, "+0.25, 0.25-0.15 and 0.15-0.074" changed to "+0.25, 0.25-0.15, and 0.15-0.074".

Page 8, line 52, "for 2,3,4,5 and 6" changed to "for 2,3,4,5, and 6".

Page 10, "Table 4 Measurement results of main elements in graphite%" changed to "Table 4 Measurement results of main elements in graphite (%)".

Page 11, line 8, "treatmentflotation" changed to "treatment flotation".

Page 11, line 10, "flotationwas" changed to "flotation was".